# OMNIPORTRAIT: FINE-GRAINED PERSONALIZED PORTRAIT SYNTHESIS VIA PIVOTAL OPTIMIZATION

**Dongxu Yue[1], Bo Lin[2], Yao Tang[2], Jiajun Liang[2], Zhihai He[3*], Chun Yuan[1*]**
[1]Tsinghua University, China    [2]JIIOV Technology, China
[3]Southern University of Science and Technology, China
`ydx25@mails.tsinghua.edu.cn`

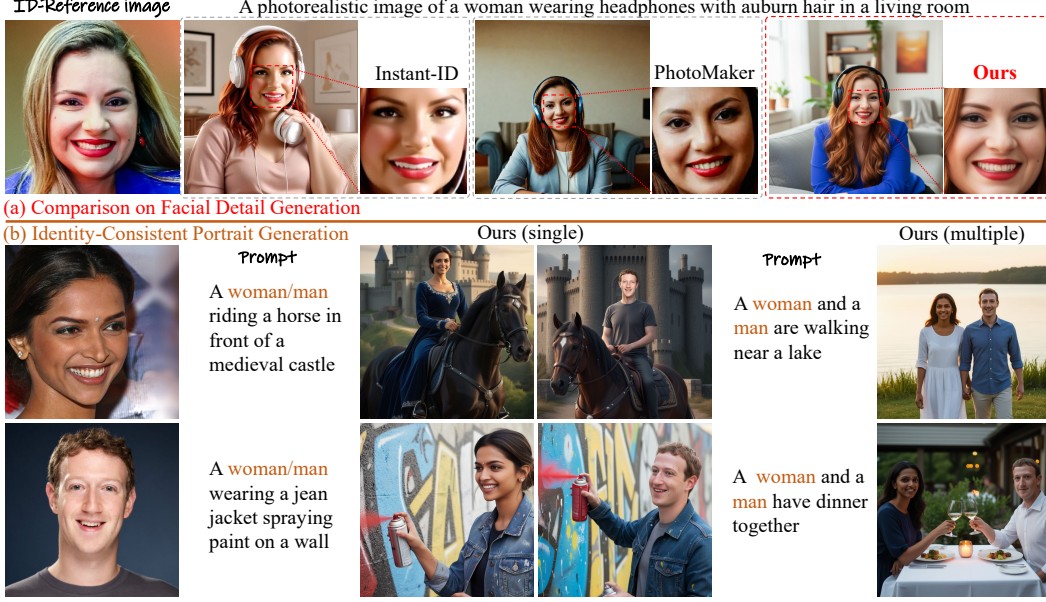

Figure 1: We demonstrate (a) the challenges faced by existing methods in customizing fine-grained facial features, and (b) that *OmniPortrait* is able to generate images with exceptional identity similarity, strong text–image alignment and can be extended to multi-identity customization.

## ABSTRACT

Image identity customization aims to synthesize realistic and diverse portraits of a specified identity, given a reference image and a text prompt. This task presents two key challenges: (1) generating realistic portraits that preserve fine-grained facial details of the reference identity, and (2) maintaining identity consistency while achieving strong alignment with the text prompt. Our findings suggest that existing single-stream methods fail to capture and guide fine-grained identity details. To address these challenges, we introduce *OmniPortrait*, a novel diffusion-based framework for fine-grained identity fidelity and high editability in portrait synthesis. Our core idea is pivotal optimization, which leverages dual-stream identity guidance in a coarse-to-fine manner. First, a Pivot ID Encoder is proposed and trained with a face localization loss while avoiding the degradation of editability typically caused by fine-tuning the denoiser. Although this encoder primarily guides coarse-level identity synthesis, it provides a good initialization that serves as the identity pivot for optimization during inference. Second, we propose Reference-Based Guidance, which performs on-the-fly feature matching and optimization over diffusion intermediate features conditioned on the identity pivot. In addition, our approach is able to generalize naturally to multi-identity customized

---

*Corresponding authors.

image generation scenarios. Extensive experiments demonstrate significant improvements in both identity preservation and text alignment, establishing a new benchmark for image identity customization.

# 1 INTRODUCTION

Recent text-to-image generation models Xu et al. (2018); Dhariwal & Nichol (2021); Rombach et al. (2022); Nichol et al. (2022); Yu et al. (2025b); Saharia et al. (2022); Li et al. (2024a), especially diffusion models Ho et al. (2020); Song et al. (2021); Yue et al. (2023); Chen et al. (2023a); Esser et al. (2024); Guo & Lin (2024), have made significant developments. Stemming from T2I diffusion models, personalized portrait synthesis task is achieved by injecting identity conditions into the conditional space of the base model, enabling the synthesis of realistic and diverse portraits for a specified identity. Recent advances in customized generation have explored two main directions: test-time fine-tuning Ruiz et al. (2023); Gal et al. (2022); Kumari et al. (2023), and design an additional encoders in the conditional space of diffusion models Gal et al. (2023); Wei et al. (2023); Arar et al. (2023); Valevski et al. (2023); Chen et al. (2023b); Ma et al. (2023a); Yue et al. (2024).

Despite significant advancements in customized generation, two major challenges remain for synthesizing identity-personalized portraits: (1) generating realistic portraits that faithfully preserve the fine-grained facial details of the reference identity, and (2) maintaining strong identity consistency while ensuring precise alignment with the text prompt. As illustrated in Fig. 1 (a), existing state-of-the-art methods Wang et al. (2024); Li et al. (2024b) struggle to faithfully preserve the facial details of the reference identity image. The loss of critical facial details, such as beauty marks, may give the impression of an overly retouched or even fake photograph, thereby hindering their practical applicability. Recent works, such as FastComposer Xiao et al. (2023), attempt to enforce the presence of the reference portrait by full fine-tuning. However, it damages the rich priors of pretrained diffusion models and often leads to imbalanced visual fidelity and editing flexibility, as illustrated in Fig. 2.

Inspired by PTI Roich et al. (2022), our core idea differs from existing single-stream conditioning methods by introducing pivotal optimization, which leverages dual-stream identity guidance in a coarse-to-fine manner. We start by introducing a Pivot ID Encoder, a vision encoder that takes the reference identity image as input and injects identity features into the conditional space of the denoising network, thereby establishing a pivot for portrait generation. To train the Pivot ID Encoder, we design a face localization loss that encourages the identity embedding of the reference image to focus on the facial region. This not only enhances identity similarity but also enables more accurate localization face during inference. Throughout training, the parameters of the base model remain frozen to ensure text alignment capability is preserved without degradation.

Although the Pivot ID Encoder alone provides only coarse-grained identity consistency, it establishes a strong initialization that serves as the identity pivot for optimization. Building on this, we introduce Reference-Based Guidance (RB-Guidance), which performs feature-space matching between the diffusion intermediate features of the reference identity image and those of the generated image, thereby enabling pivotal optimization guided by identity features. Thanks to the Pivot ID Encoder design, the rich priors and text alignment capabilities of the base model are preserved, while identity and facial details from the reference image are guaranteed through RB-Guidance. Moreover, OmniPortrait is plug-and-play and is readily compatible with many other methods or plugins, while naturally extending to multi-identity customization scenarios, as shown in Fig. 1 (b).

Equipped with these techniques, as shown in Fig. 1, OmniPortrait demonstrates extraordinary customization capabilities with just a single reference image and achieves state-of-the-art results, showing a significant quality advantage over prior works in similar settings. In summary, our contributions are as follows:

- We propose OmniPortrait, a pivotal optimization based framework for detail-preserving identity-customized portrait synthesis. It incorporates Pivot ID Encoder and Reference-Based Guidance to enable coarse-to-fine personalized portrait generation, enhancing identity similarity with minimal impact on generative capability.
- We design Reference-Based Guidance (RB-Guidance), a novel test-time optimization framework based on local diffusion feature matching, which enables training-free preservation of fine-grained identity details.

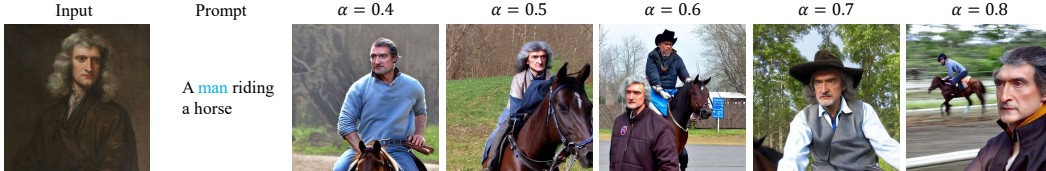

Figure 2: The trade-off between identity fidelity and text editability in FastComposer Xiao et al. (2023) controlled by its delay condition parameter $\alpha$.

- We construct a large-scale human face dataset with detailed annotations, containing 1 million data pairs, which can facilitate training for identity-customized image generation tasks.

- Extensive experiments demonstrate the excellent performance of our framework particularly in preserving fine-grained facial details, Results show that our method achieves superior identity fidelity without compromising editability.

## 2 RELATED WORK

**Text-to-Image Diffusion Models** In recent years, diffusion models Ho et al. (2020); Nichol & Dhariwal (2021) have rapidly advanced in image generation, with models like Stable Diffusion Rombach et al. (2022), DALL-E2, and Imagen Rombach et al. (2022); Saharia et al. (2022) producing high-quality images. More recent models Peebles & Xie (2023); Yu et al. (2025a); Chen et al. (2023a); Guo et al. (2025); Esser et al. (2024) adopt Transformer-based denoising networks and leverage larger datasets, resulting in stronger generalization and enhanced image generation.

**Few-shot Finetuning Customized Generation** Subject-driven image generation enables models to create images with personalized content. Some methods Ruiz et al. (2023); Gal et al. (2022); Kumari et al. (2023) fine-tune models on multiple reference images to capture a specific subject or face. However, these approaches are constrained by the requirement for several images of the same subject and the high computational costs Wang et al. (2025a); Jo et al. (2025).

**Encoder-Based Customized Generation** Recent works Gal et al. (2023); Wei et al. (2023); Ma et al. (2023b); Shi et al. (2023); Arar et al. (2023); Ye et al. (2023); Xiao et al. (2023); Guo et al. (2024); Qian et al. (2025); Mou et al. (2025); Jiang et al. (2025); Huang et al. (2024) use visual encoders for efficient image synthesis but struggle with balancing fidelity and editing flexibility. Building on reference image injection paradigm, subsequent studies have explored enhancements in multiple directions, including improved preservation of facial identity Li et al. (2024b) and extended support for multiple reference subjects Gu et al. (2023); Kwon & Ye (2024). InstantID Wang et al. (2024) enhances identity-consistent generation through a dedicated face encoder and identity network.

**Training-free Consistent Generation** Training-free approaches have recently attracted significant attention owing to their computational efficiency. For instance, MasaCtrl Cao et al. (2023) proposed mutual self-attention, in which the keys and values of self-attention are substituted with those derived from the reference image. Similarly, ConsiStory Tewel et al. (2024) introduced subject driven self-attention, allowing each frame to reference multiple subjects from different images in a batch. They further incorporated DIFT Tang et al. (2023) for feature injection in self-attention to improve detail consistency. Despite these advances, existing methods still struggle to preserve the fine facial details of reference images, which remains a challenging problem worthy of further investigation.

## 3 METHOD

Given an identity reference image $x_{ref}$ and a target prompt $P_t$ that specifies the desired scene, the objective is to generate an image aligned with the target prompt while preserving the fine-grained identity details of the reference face. Ideally, customized portrait generation should achieve high fidelity in identity preservation while simultaneously maintaining satisfactory alignment with the target prompt $P_t$. To this end, OmniPortrait is introduced, which is built upon latent diffusion model and extended from energy-based diffusion guidance. Through Pivot ID Encoder and RB-Guidance, customized portraits are generated in a coarse-to-fine manner, the overview is shown in Fig. 3.

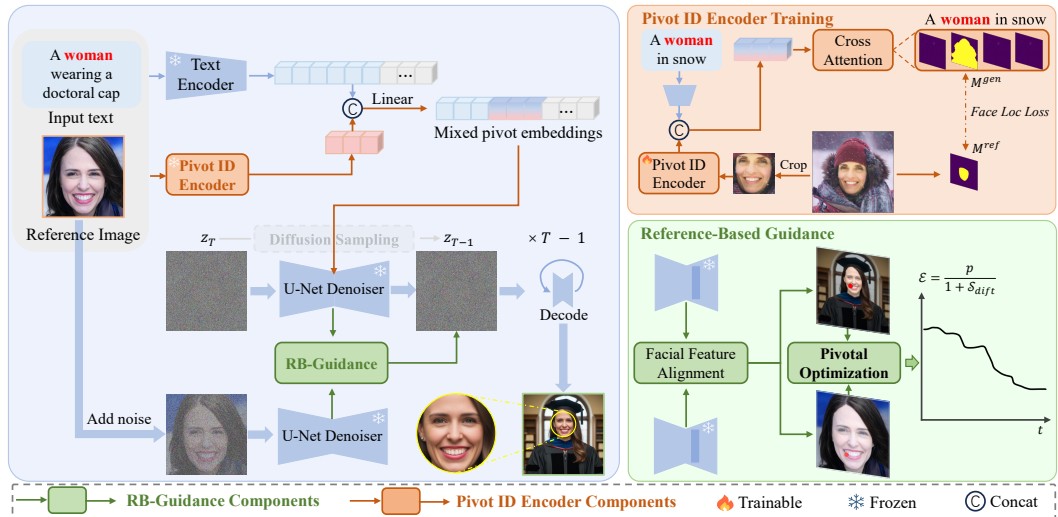

Figure 3: **Overviews of the proposed *OmniPortrait*, Pivot ID Encoder training pipeline and Reference-Based Guidance**. We train the proposed Pivot ID Encoder with a face localization loss. During inference, the Pivot ID Encoder is kept frozen and injects identity information from the reference image into the denoiser to obtain the identity pivot, In addition, RB-Guidance is introduced to optimize an energy function, thereby enhancing the preservation of facial identity features.

## 3.1 PRELIMINARIES

**Latent Diffusion Model Sampling with Classifier-free Guidance** The latent diffusion model is a type of generative model that reduces images to a low-dimensional latent representation using an autoencoder Kingma & Welling (2013); Wang et al. (2025b) and then iteratively denoise in the latent space. DDIM sampling is a deterministic and non-Markovian process that generates image latents by progressively denoising in the latent space. Formally, the DDIM sampling procedure can be divided into two components: a denoising step based on Tweedie's formula and a subsequent re-noising step. Given a noisy latent $z_t \in \mathbb{R}^{h \times w \times c}$ and a text condition $\mathbf{y}$, the update rule of DDIM can be expressed as:

$$z_{t-1} = \sqrt{\bar{\alpha}_{t-1}} \hat{z} \left[ \epsilon_\theta \left( z_t, t, \mathbf{y} \right) \right] + \sqrt{1 - \bar{\alpha}_{t-1}} \epsilon_\theta \left( z_t, t, \mathbf{y} \right), \tag{1}$$

$$\hat{z} \left[ \epsilon_\theta \left( z_t, t, \mathbf{y} \right) \right] := \left( z_t - \sqrt{1 - \bar{\alpha}_t} \epsilon_\theta \left( z_t, t, \mathbf{y} \right) \right) / \sqrt{\bar{\alpha}_t}, \tag{2}$$

where a denoiser is used to predict noise $\epsilon_\theta \left( z_t, t, \mathbf{y} \right)$ from the noisy latent $z_t$ at a specified time step $t$, given text condition $\mathbf{y}$, where $\alpha_t$ and $\sigma_t$ are predefined functions of $t$ that determine the diffusion process. In addition, $\alpha_t = 1 - \beta_t$ and $\bar{\alpha}_t = \prod_{i=1}^{t} \alpha_i$ in which $\beta_t \in (0, 1)$ is a variance schedule.

Classifier-free guidance is employed to enhance the controllability of diffusion models. Accordingly, during training, the model is required to randomly drop the condition $\mathbf{y}$ in $\epsilon_\theta \left( z_t, t, \mathbf{y} \right)$ with a certain probability. During inference, the predicted noise are formulated as follows:

$$\hat{\epsilon}_{\mathbf{y}} \left( z_t \right) = \epsilon_\theta \left( z_t, t, \varnothing \right) + w \left( \epsilon_\theta \left( z_t, t, \mathbf{y} \right) - \epsilon_\theta \left( z_t, t, \varnothing \right) \right), \tag{3}$$

where $w > 1$ denotes the guidance scale, and $\epsilon_\theta \left( z_t, t, \varnothing \right)$ is the unconditional prediction.

**Energy Diffusion Guidance** From the continuous perspective of score-based diffusion, condition $\mathbf{y}$ can be combined by a score function $\nabla_{z_t} \log p \left( z_t \mid \mathbf{y} \right)$ and can be further decomposed as:

$$\nabla_{z_t} \log q \left( z_t \mid \mathbf{y} \right) = \log \left( \frac{q \left( \mathbf{y} \mid z_t \right) q \left( z_t \right)}{q(\mathbf{y})} \right) \propto \nabla_{z_t} \log q \left( z_t \right) + \nabla_{z_t} \log q \left( \mathbf{y} \mid z_t \right), \tag{4}$$

where $\nabla_{z_t} \log q \left( z_t \right)$ corresponding to $\epsilon_\theta \left( z_t, t, \varnothing \right)$ and $q \left( \mathbf{y} \mid z_t \right) \propto \mathcal{E} \left( z_t; t, \mathbf{y} \right)$. $\mathcal{E}$ is an energy function and it usually requires an additional model to estimate the conditional probability between the noisy latent $z_t$ and the condition $\mathbf{y}$.

## 3.2 PERSONALIZED IDENTITY PIVOT

**Pivot ID Encoder** As shown in Figure 3, Our goal is to associate the identity contained in the reference image with a specific text token, such as *woman* or *man*, that most effectively represents the desired identity concept. To achieve this, A Pivot ID Encoder is deployed to extract visual features $e_{ref}$ from the reference image $x_{ref}$. To bridge the gap between visual embedding $e_{ref}$ and text embedding $e_{txt}$, a linear projection layer is adopted as a feature alignment network. Specifically, we concatenate $e_{ref}$ and text embedding $e_{txt}$ before feeding them into the linear projection layer, and then we get $e_{mix}$ as mixed pivot embeddings.

**Training with Face Localization Loss** Previous practices have attempted to achieve high-fidelity face preservation by fine-tuning the denoising network. However, our observations indicate that such training compromises the priors embedded in the pre-trained model, which in turn leads to abnormal scene composition and poor editability, as illustrated in Fig. 2. Therefore, in this work, only the Pivot ID Encoder and a linear projection layer are trained, which not only enables efficient fine-tuning but also makes the Pivot ID Encoder plug-and-play.

Specifically, we employ an LLM Bai et al. (2023) to extract person-related nouns from the image captions. A face detection model and a face parsing model are used to obtain cropped face images and corresponding face masks $M$. These elements are then combined to construct triplets consisting of a caption noun, a cropped face image and a face mask. More details are presented in Section 4.1.

To obtain an accurate facial region for subsequent pivotal optimization, it is necessary to ensure that the facial mask of the generated portrait is available at the start of sampling. A straightforward way is to derive this mask from the cross-attention map of $e_{mix}$. However, our findings suggest that when the Pivot ID Encoder is trained merely with diffusion loss, the cross-attention of $e_{mix}$ tends to spread across the entire person rather than being restricted to the facial region. To address this issue, we propose a face localization loss to constrain the attention region of $e_{mix}$. Specifically, let $A_t \in [0,1]^{(h \times w) \times n}$ be the cross-attention map in sampling timestep $t$, where $A[i,j,k]$ denotes the information flow from the $k$-th conditional token to the $(i,j)$ latent pixel. Let $m$ be the index of $e_{mix}$, and $A_t(m) = A[:,:,m] \in [0,1]^{(h \times w)}$ be the cross-attention map of the mixed pivot embeddings. We supervise the cross-attention map $A_m$ to be close to the resized segmentation mask $M$, and the overall training objective can be formulated as:

$$\mathcal{L} = \mathcal{L}_{diff} + \eta \mathcal{L}_{loc}$$

$$= \mathbb{E}_{t \sim \mathcal{U}(1,T), \epsilon_t \sim \mathcal{N}(0,I)} \left[ \| \epsilon_t - \epsilon_\theta \left( z_t, t, C \right) \|^2 \right] + \frac{\eta}{hw} \sum_{i=1}^{h} \sum_{j=1}^{w} \left( A_t(i,j) - M(i,j) \right)^2, \quad (5)$$

where $\eta = 2e{-}3$ is a hyperparameter, and we apply $\mathcal{L}_{loc}$ at $A_t(m) \in [0,1]^{(16 \times 16)}$.

## 3.3 REFERENCE BASED GUIDANCE

**Regional Mask Extraction** By training the Pivot ID Encoder with the proposed facial localization loss, the cross-attention map $A_t^c(m)$ corresponding to $e_{mix}$ can be leveraged during inference to obtain the facial mask of the generated image. However, directly applying a threshold to $A_t^c(m)$ can produce a coarse-grained mask. We choose to incorporate a self-attention map rich in structural information, and design a novel iterative refinement procedure, which can be formally expressed as:

$$\hat{A}_t^c(m) = A_s(m) \cdot \underbrace{\text{norm} \left( \text{norm} \left( \cdots \text{norm} \left( A_t^c(m) \right)^\alpha \cdots \right)^\alpha \right)^\alpha}_{\gamma \text{ times}}. \quad (6)$$

Here, norm refers to the min–max normalization to adjust the map value fall within the range $[0,1]$. After $\gamma = 3$ iterations with $\alpha = 2$, a threshold $\beta = 0.5$ is then applied to obtain the mask $M_{gen}$.

**Diffusion Feature Correspondence** The human face in a portrait contains rich fine-grained details, which serve as the implicit basis for human perception of identity. Therefore, our goal is to devise an energy guidance that explicitly focuses on the fine-grained facial features. Recent works Tang et al. (2023) have shown strong local correspondence between intermediate features in diffusion models , which can be used for dense matching between different images. During the sampling process, the diffusion intermediate feature $D_t^{gen}$ enables the noisy latent code $z_t$ at timestep $t$ to be directly

mapped through the denoising network. Similarly, we can extract $D_{t_0}^{ref}$ from the reference image $x_{ref}$ by adding $t_0$ steps noise, and we set $t_0 = 671$ in practice. Before the sampling process, a face parsing model is employed to extract the facial mask $M^{ref}$ from the reference image $x_{ref}$. We first identify, within the feature spaces $D_{t_0}^{ref}$ and $D_t^{gen}$, the set of best-matching points $p_{gen}$ within the mask $M^{gen}$ that correspond to the points $p_{ref}$ located inside the reference mask $M^{ref}$:

$$p_{gen} = \underset{p \in M^{ref}}{\operatorname{argmin}} d\left( D_{t_0}^{ref}\left[p_{ref}\right], D_t^{gen}[p]\right), \tag{7}$$

where $d(\cdot, \cdot)$ donates Euclidean distance. We then measuring correspondence by:

$$\mathcal{S}_{dift} = \sum_{p_{ref} \in M^{ref}} \frac{\cos\left(D_t^{gen}\left[p_{gen}\right], D_{t_0}^{ref}\left[p_{ref}\right]\right) + 1}{2}. \tag{8}$$

**Background Gradient Masking** We aim to make the face details in $x_{ref}$ closely resemble the one in the generated image. Thus, our optimization objective is to maximize the similarity $\mathcal{S}_{dift}\left(z_t, x_{ref}, M^{gen}, M^{ref}\right)$. The gradient has the same dimensionality as $z_t$, allowing pixel-wise optimization. Nevertheless, this causes both the foreground and background to be modified simultaneously, so regions unrelated to the face are also influenced by the gradient, leading to image blurring. To address this issue, we apply mask $M^{gen}$ on Equation. 8 to filter out the gradients corresponding to the background before performing the guidance step:

$$\hat{\mathcal{S}}_{dift} = \sum_{p_{ref} \in M^{ref}} \frac{\cos\left(D_t^{gen}\left[p_{gen}\right], D_{t_0}^{ref}\left[p_{ref}\right]\right) + 1}{2} \odot M^{gen}, \tag{9}$$

**On the Fly Pivotal Optimization** During inference, the trained Pivot ID Encoder takes a single reference image as input. Although it is difficult to fully preserve the fine-grained facial details of the reference, it provides a strong initialization and accurately localizes the facial region of the generated portrait. In practice, during the early denoising stages, the generated image only contains coarse-level concepts. This makes it challenging to establish precise matches between the reference image and the generated image, which in turn causes the guidance gradients of RB-Guidance to diverge. To address this issue, pivotal optimization is introduced starting from timestep $\hat{t} = uT$ and $u = 0.6$, which achieves the best trade-off between guidance effectiveness and stability, as illustrated in Fig. 7. Accordingly, We further formulate diffusion feature correspondence as an energy function:

$$\mathcal{E} = \frac{p}{1 + \hat{\mathcal{S}}_{dift}\left(z_t, x_{ref}, M^{gen}, M^{ref}\right)}, \tag{10}$$

where $p$ denotes the hyperparameter that controls the strength of pivotal optimization and the pivotal optimization can be formulated as follows:

$$\hat{\boldsymbol{\epsilon}}_{\mathbf{y}}\left(\boldsymbol{z}_t\right) = \begin{cases} \boldsymbol{\epsilon}_\theta\left(\boldsymbol{z}_t, t, \varnothing\right) + w\left(\boldsymbol{\epsilon}_\theta\left(\boldsymbol{z}_t, t, \mathbf{y}\right) - \boldsymbol{\epsilon}_\theta\left(\boldsymbol{z}_t, t, \varnothing\right)\right) & \text{if } t > \hat{t} \\ \boldsymbol{\epsilon}_\theta\left(\boldsymbol{z}_t, t, \varnothing\right) + w\left(\boldsymbol{\epsilon}_\theta\left(\boldsymbol{z}_t, t, \mathbf{y}\right) - \boldsymbol{\epsilon}_\theta\left(\boldsymbol{z}_t, t, \varnothing\right)\right) + \nabla_{z_t}\mathcal{E} & \text{if } t <= \hat{t} \end{cases} \tag{11}$$

## 4 EXPERIMENT

### 4.1 DATASET CONSTRUCTION

Although customized portrait generation has been extensively studied, previous efforts typically relied on private datasets for training Wang et al. (2024); Li et al. (2024b). Moreover, most existing multimodal face datasets only guarantee the existence of human faces within the images, while lacking precise annotations of individual body positions and facial regions. Motivated by these limitations, we construct and release *OmniPortrait-1M*, a large-scale, high-quality multimodal face dataset with detailed annotations, which can support a variety of personalized portrait generation tasks. We collect raw data from Pexels, COYO-700M Byeon et al. (2022) and LAION-2B Schuhmann et al. (2022), and further filtered by resolution and aesthetic scores. To annotate faces, YOLOv7-Face [1] is used to detect all faces, after which images containing no faces or multiple faces are discarded. In addition, YOLOX Ge et al. (2021) is employed to generate bounding boxes for each person, and BLIP-2 Li et al. (2023) is used to produce image caption. Details are in the **Appendix**.

---

[1] https://github.com/derronqi/yolov7-face

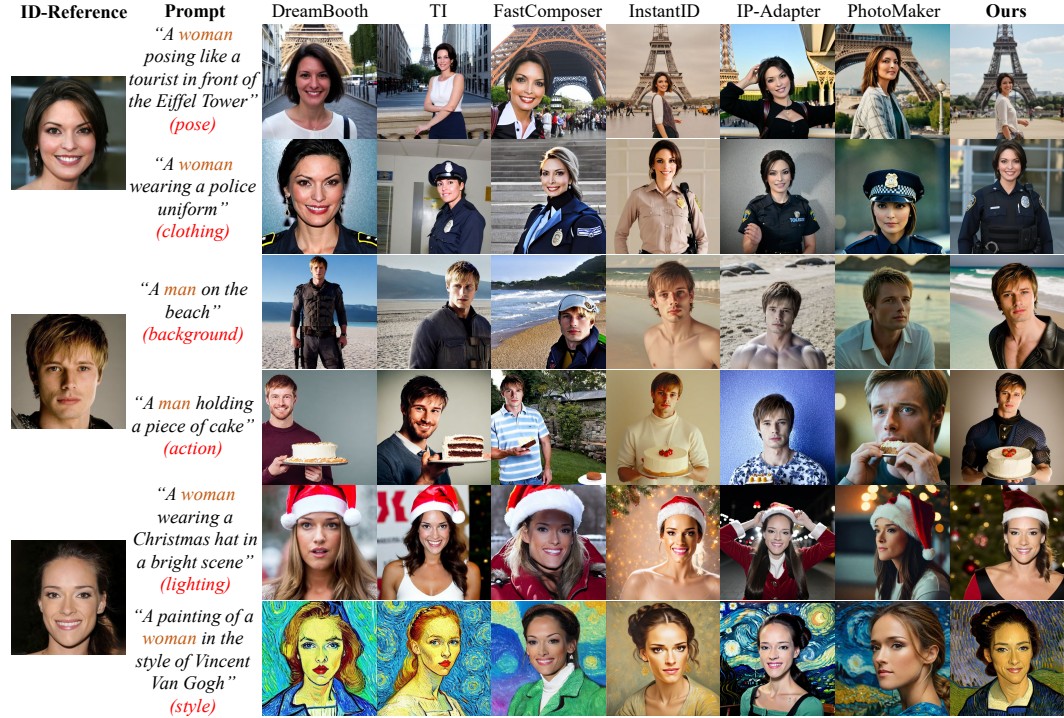

Figure 4: **Qualitative comparisons**. We compare our method with baseline approaches in terms of text alignment across several aspects, including pose, clothing, background, action, lighting, and style. OmniPortrait achieves the best text editability while effectively preserves the identity.

## 4.2 EXPERIMENTAL SETUP

**Implementation Details** The pretrained backbones utilized in our experiments include SD Rombach et al. (2022) and SDXL Podell et al. (2023) based community models. For the Pivot ID Encoder, OpenCLIP-ViT-L/14 Cherti et al. (2023) is adopted. We train of the Pivot ID Encoder and the linear projection layer using the AdamW optimizer with a learning rate of $2e-5$. SD-based Realistic Vision is optimized for 80k steps on four NVIDIA A100 80GB GPUs. For SDXL-based RealVisXL, images are filtered to 1024 resolution, followed by 60k training steps. To enable classifier-free guidance, we randomly drop the text and ID condition with a 10% probability. During inference, $w$ is set to 7.5 and $T = 1000$ with 50 sampling steps, and the strengths of RB-Guidance $p$ is set to $8.5$. We conduct the subsequent experiments using SDXL.

**Evaluation Settings** For evaluation, 50 reference images are sampled from the CelebA-HQ Karras et al. (2017) and FFHQ Karras et al. (2019) datasets, and 30 text prompts are constructed for each identity. To comprehensively assess the effectiveness of the proposed OmniPoratrait, three categories of metrics are considered: (1)*Text Editability*. CLIP-T Hessel et al. (2021) and BLIP scores are employed to measure the consistency between the input text prompts and the generated results. (2)*Identity Fidelity*. We report CLIP-I scores between the reference face image and the generated image. In addition, we adopt RetinaFace Deng et al. (2020) as the detection model, SIM represents the ID cosine similarity, with ID embeddings extracted by FaceNet Schroff et al. (2015). (3)*Image Quality*. CLIP-IQA and FID Heusel et al. (2017) are adopted to evaluate perceptual quality.

## 4.3 QUALITATIVE COMPARISON

We compare with finetuning-based methods, including DreamBooth Ruiz et al. (2023) and Textual Inversion Gal et al. (2022), as well as encoder-based methods, including Fastcomposer Xiao et al. (2023), InstantID Wang et al. (2024), Photomaker Li et al. (2024b) and IP-Adapter Ye et al. (2023). Note that we use the IP-Adapter-FaceID-Plus version for a fair comparison. We test each well-prepared model on a set of challenging prompts. Fig. 4 shows the generated samples from our method and the baseline methods. It demonstrates that our OmniPortrait exhibits higher text-image

Table 1: **Quantitative comparisons.** "Plug and Play" refers to whether the method supports control conditions beyond text, such as pose keypoints and segmentation maps. Our method achieves optimal performance in both text alignment and identity preservation.

| Method | Plug and Play | Text Editability | | Identity Fidelity | | Image Quality | |
|---|---|---|---|---|---|---|---|
| | | BLIP↑ | CLIP-T↑ | CLIP-I↑ | SIM↑ | IQA↑ | FID↓ |
| DreamBooth | ✗ | 78.42 | 23.32 | 66.45 | 60.07 | 85.88 | 219.40 |
| Textual Inversion | ✗ | 76.36 | 23.89 | 58.32 | 58.32 | 71.73 | 235.57 |
| FastComposer | ✗ | 78.87 | 22.98 | 69.83 | 64.97 | 82.04 | 223.94 |
| PhotoMaker | ✗ | **80.51** | 23.97 | 67.64 | 62.68 | 70.51 | 236.93 |
| InstantID | ✓ | 78.21 | 22.26 | 73.03 | 68.35 | 84.17 | 221.62 |
| IP-Adapter | ✓ | 79.33 | 23.93 | 68.23 | 64.19 | **88.15** | **211.15** |
| OmniPortrait (Ours) | ✓ | 80.24 | **24.25** | **73.08** | **69.13** | 86.80 | 213.48 |

alignment while capturing the face details of the identity in the reference face. IP-Adapter and FastComposer achieve relatively high facial fidelity but at the cost of poor text alignment. Their results often resemble a copy-and-paste of the reference image, as shown in the last row of Fig. 4.

## 4.4 QUANTITATIVE COMPARISON

**Metric Evaluation** Despite our method requiring only a single reference image, to prevent catastrophic overfitting of our baselines, we provide 5 images for each ID to optimization-based methods like DreamBooth. As shown in Table 1, OmniPortrait outperforms all baselines in both identity preservation and text-alignment metrics, indicating better text control capability while preserving the identity of the reference image. To further compare the performance of our method and the baselines in terms of text alignment and identity fidelity, we conduct five sets of experiments for each method within their tunable parameter ranges, where each set uses the same reference identity image and prompt. As shown in Fig. 5, baseline methods, which rely on fine-tuning or single-stream identity encoding, struggle to strike a balance between prompt consistency and identity preservation. In contrast, OmniPortrait achieves the best overall performance.

**User Study** We conducted a user study to compare the proposed approach against baseline methods. Thirty participants from diverse backgrounds were recruited to evaluate 20 sets of generated images. Each set consisted of one reference identity image and five textual prompts, with every baseline method producing five corresponding predictions. Participants were asked to select the top-2 best-performing method according to four aspects: text alignment, identity fidelity, facial detail preservation, and the overall naturalness of the generated results. We present the results in the **Appendix**, the proposed OmniPortrait is consistently preferred over the baseline methods across all evaluation criteria. The advantage is particularly pronounced in facial detail preservation, where more than 50% of participants

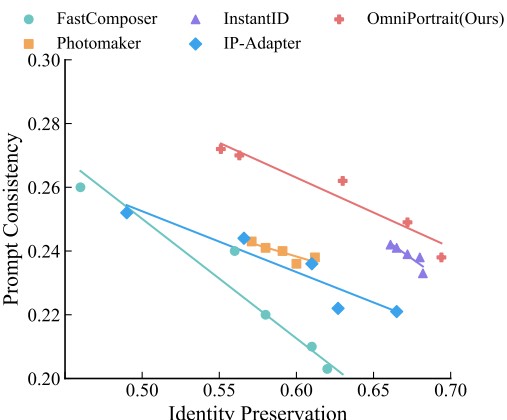

Figure 5: Comparisons on prompt consistency and identity pereservation.

judged the generated portraits to exhibit a closer resemblance to the reference identity image. These results highlight the effectiveness of OmniPortrait in achieving a superior balance between semantic editability and faithful identity preservation.

## 4.5 ABLATION STUDY

**Pivot ID Encoder with Face Localization Loss** We first perform an ablation study on the face localization loss $\mathcal{L}_{loc}$ used for training the Pivot ID Encoder (PIE). As illustrated in Fig 6, removing the face localization loss results in inaccurate ID inject regions of the mixed pivot embeddings $e_{mix}$ during inference. This misalignment further causes partial mismatches between the diffusion

features of the reference face and those of the generated subject, which leads to degraded identity fidelity and corruption of regions unrelated to the face as shown in Tab. 2.

Our core idea is to employ the pivot embeddings obtained by encoding the reference face image through the Pivot ID Encoder, which serve as a reliable initialization for guidance during inference. Without such embeddings, the optimization based on the reference image cannot properly localize the target region or establish feature correspondences. Consequently, the guidance gradients are propagated to non-pivot areas, resulting in no identity enhancement as shown in Fig 6 and Tab. 2.

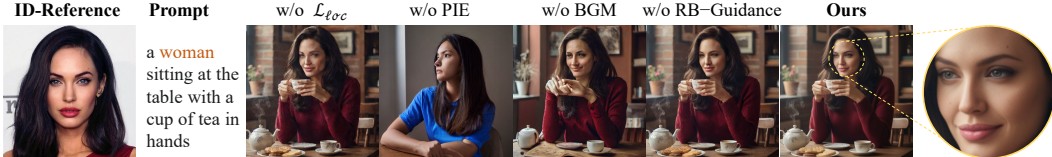

Figure 6: Qualitative ablation study on OmniPortrait.

**Background Gradient Masking** As shown in Fig 6, when the guidance gradients are applied globally across the image, the effect of aligning the matched feature points between the reference and generated faces is weakened, making it challenge to achieve high-fidelity preservation of facial details. We attribute this phenomenon to gradient leakage, where the diffusion features of the reference face become implicitly coupled with the non-facial regions of the generated image.

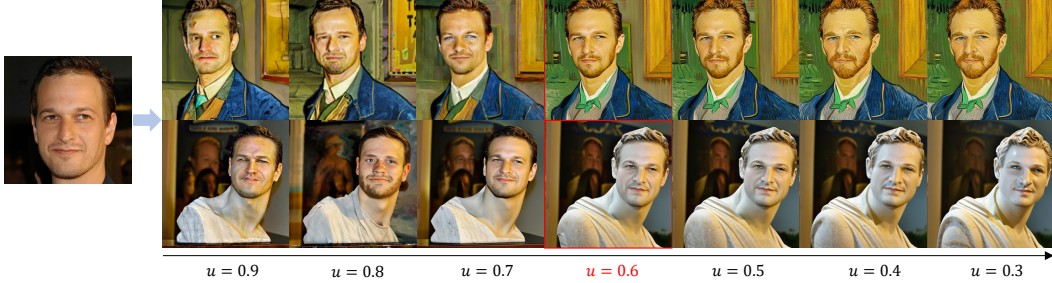

Figure 7: Qualitative ablation study on starting time parameter introduced by RB-Guidance.

**Pivotal Optimization** An ablation study was first conducted on the starting time parameter $\hat{t} = uT$ introduced by RB-Guidance. As shown in Fig 7, during the early denoising stages, the generated image contains only coarse-level concepts, making it difficult to establish precise correspondences between the diffusion features of the reference image and those of the generated image, which causes the guidance gradients of RB-Guidance to diverge. Conversely, applying the guidance at a later stage reduces its effectiveness. Therefore, $u = 0.6$ was selected in our experiments. As illustrated in Fig 6, in the absence of RB-Guidance, the generation process is conditioned solely on the mixed pivot embeddings encoded by the Pivot ID Encoder, and the resulting portraits exhibit low identity similarity and insufficient preservation of facial details.

Table 2: Quantitative ablation study on OmniPortrait.

| $\mathcal{L}_{loc}$ | PIE | BGM | RB-Guidance | BLIP↑ | CLIP-T↑ | CLIP-I↑ | SIM↑ | IQA↑ | FID↓ |
|---|---|---|---|---|---|---|---|---|---|
| ✗ | ✓ | ✓ | ✓ | 77.32 | 22.11 | 46.54 | 35.01 | 68.10 | 476.22 |
| ✓ | ✗ | ✓ | ✓ | 75.78 | 23.19 | 21.83 | 14.15 | 63.98 | 383.17 |
| ✓ | ✓ | ✗ | ✓ | 66.80 | 19.34 | 37.30 | 33.42 | 52.44 | 483.93 |
| ✓ | ✓ | ✓ | ✗ | **81.05** | **24.88** | 66.10 | 63.87 | 85.51 | **210.45** |
| ✓ | ✓ | ✓ | ✓ | 80.24 | 24.25 | **73.08** | **69.13** | **86.80** | 213.48 |

## 4.6 EXTENDING TO MULTI-IDENTITY PERSONALIZED PORTRAIT SYNTHESIS

For multi-identity personalized portrait synthesis, we instantiate a separate Pivot ID Encoder per reference identity to obtain its pivot embedding. Each embedding is concatenated with the corresponding person-related text tokens such as woman or man, and RB-Guidance is then applied in the respective face spatial regions. This design naturally extends our pipeline to multi-identity customized generation, as shown in Fig. 1. More corresponding results are provided in the **Appendix**.

## 5 CONCLUSION

We presented OmniPortrait, a diffusion-based framework for identity-preserved portrait synthesis that addresses the core limitations of prior single-stream approaches—namely insufficient fine-grained identity fidelity and degraded text alignment. Central to our design is pivotal optimization: a Pivot ID Encoder, trained with a face-localization loss, establishes a robust identity pivot without fine-tuning the base model, while Reference-Based Guidance (RB-Guidance) performs test-time, coarse-to-fine optimization via local matching of diffusion intermediate features. This dual-stream strategy preserves the rich priors and editability of the pretrained backbone, enabling high-quality, identity-faithful, and prompt-consistent generation. To further facilitate research on personalized portrait synthesis, we will open-source the large-scale human-face dataset constructed in this work. Extensive experiments demonstrate that OmniPortrait achieves state-of-the-art performance on identity similarity and text–image alignment, with clear qualitative gains in fine facial details. The framework is plug-and-play, readily compatible with community models and naturally extensible to multi-identity scenarios, offering practical value to the broader community.

## ACKNOWLEDGEMENTS

This work was supported by the National Key R&D Program of China (Grant No. 2022YFB4701400/4701402), the National Natural Science Foundation of China (Grant No. 62331014), and the SSTIC Grant (Grant Nos. KJZD20230923115106012, KJZD20230923114916032, GJHZ20240218113604008).

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

## A  APPENDIX

### A.1  DETAILS OF OMNIPORTRAIT-1M DATASET

In this section, we describe how the raw images for OmniPortrait-1M are collected and then report dataset statistics.

**Data Selection and Standardization** We collected 2.8B public raw images from Pexels, COYO-700M, and LAION-2B. We first filtered images using the criteria '*aesthetic > 5 & pwatermark < 0.5 & width > 512 & height > 512*'. In addition, we use YOLOv7-Face to detect faces and retain only single-face images, discarding those with none or multiple faces. We found that relying solely on face detection leads to false positives, including portrait-like patterns and animal faces; therefore, additional safeguards are required. To mitigate false positives, we incorporate YOLOX-based person detection and keep samples only if a person and a face are simultaneously detected and the face bounding box is spatially enclosed by the person bounding box.

The raw image files are renamed using a seven-digit number to avoid duplication. Processing was carried out with img2dataset Beaumont (2021). The standardized outputs form our raw dataset, yielding a collection of 1.16M high-quality portrait images.

**Statistics of OmniPortrait-1M** We analyze the probability distribution of the face-to-image area ratio to ensure diversity in face sizes; a higher prevalence of smaller faces promotes learning a more stable distribution, as shown in the left part of Fig. 8. We also examine the distribution of face confidence scores, as depicted in the right part of Fig. 8.

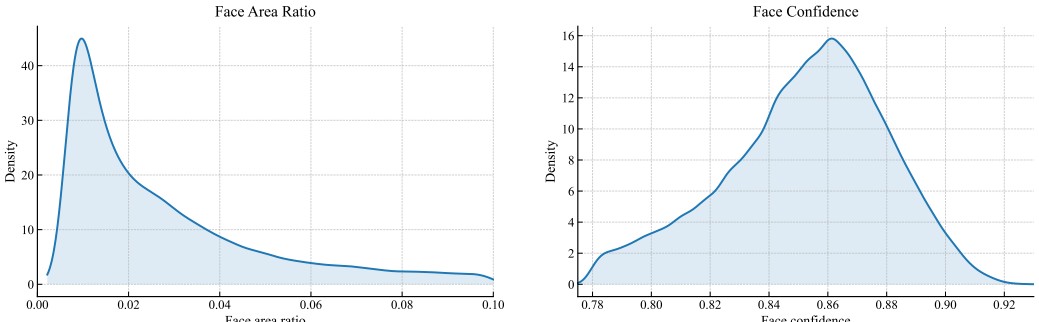

Figure 8: Distributions of face-to-image area ratios and face confidence scores in OmniPortrait-1M.

### A.2  ADDITIONAL COMPARISONS

In this section, we first present the results of our user study to reflect participants' assessments of our method and the baselines across multiple dimensions. We then provide additional qualitative comparisons against customization methods built on the FLUX backbone.

**User Study** As discussed in the main paper, a user study was conducted to more comprehensively assess OmniPortrait and baseline methods. The results are presented in Fig. 9. Our proposed OmniPortrait is preferred by the largest number of participants across four aspects: text alignment, identity fidelity, facial details, and naturalness. In particular, for facial details, over 50% of respondents selected our method as the best at preserving fine-grained facial features, demonstrating the effectiveness of the proposed dual-stream pivotal optimization.

**Comparison with FLUX-based Methods** Recent works have explored FLUX-based Labs (2024) customization by injecting identity information into the DiT backbone, either via additional conditioning encoders or token concatenation. We compare against the strongest FLUX-based methods to date, including IP-Adapter-FLUX Ye et al. (2023), PuLID-FLUX Guo et al. (2024), InfiniteYou Jiang et al. (2025), and DreamO Mou et al. (2025) (see Fig. 10). As illustrated, except for DreamO and our OmniPortrait, the first three methods tend to produce overly smooth faces, losing substantial high-frequency detail. While DreamO maintains identity fidelity, it relies on facial parsing of the reference image; this dependency leads to disharmonious blending when background defocus is applied, as shown in the last row of Fig. 10.

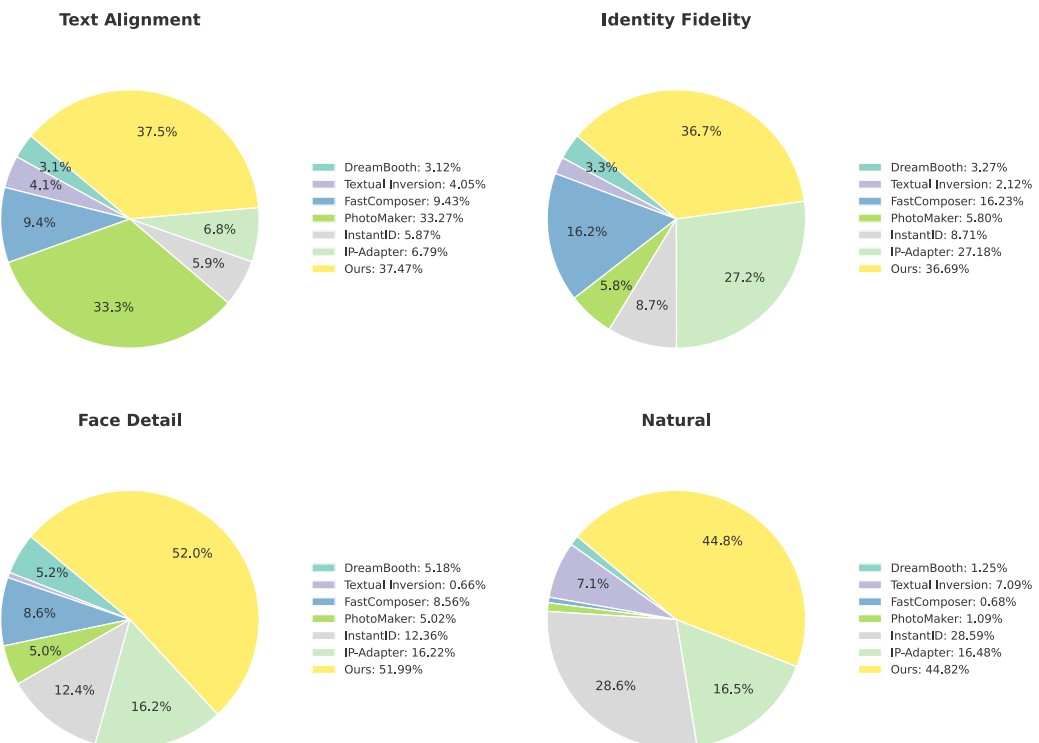

Figure 9: User study results on text alignment, identity fidelity, facial details, and naturalness.

**Extended Comparisons with Closed-Source Image Editing Models** To enable an apples-to-apples comparison, we extend Fig. 10 to evaluate closed-source image editing models under the same reference image and prompt including GPT-4o, JiMeng AI, and Nano Banana. The results are presented in Fig. 11. GPT-4o exhibits suboptimal identity fidelity; JiMeng-AI tends to copy–paste the reference face during personalization (see the second and third rows of Fig. 11); and Nano Banana often preserves more background content, reflecting a bias toward text alignment. Overall, these observations highlight the challenge of balancing identity fidelity and prompt consistency in closed-source systems.

### A.3 MORE APPLICATIONS

**Multi-Identity Personalized Portrait Synthesis** In multi-identity personalization, OmniPortrait can be adapted with minimal changes. By performing independent pivotal optimization for each identity, we simultaneously preserve per-identity consistency while applying RB-Guidance within the corresponding facial regions, thereby avoiding identity confusion. As shown in Fig. 12, we evaluate multi-identity generation under diverse text prompts; results indicate that our approach achieves strong text editability while maintaining clear, non-confounding identity consistency across multiple identities.

### A.4 ADDITIONAL ABLATIONS

**Hyperparameter Ablations** We present the hyperparameter search results in Fig. 14. regarding the face localization loss weight $\eta$ in Equation. 5, a larger $\eta$ leads to higher face similarity but compromises text alignment. The parameter $p$ in Equation. 10, which controls the strength of RB-Guidance, and $u$ in Equation. 11, which governs the guidance starting time, exhibit similar properties, and we can identify a sweet spot by balancing text alignment and face similarity. Regarding the timestep $t_0$ for the reference image, we observe that text alignment is insensitive to this parameter, whereas face similarity peaks when $t_0$ is approximately 670. A smaller $t_0$ results in more rigid feature matching, thereby impairing the structural similarity of the face. Conversely, as $t_0$ increases, the accuracy of feature matching declines rapidly, which is attributed to the intrinsic characteristics of DIFT features.

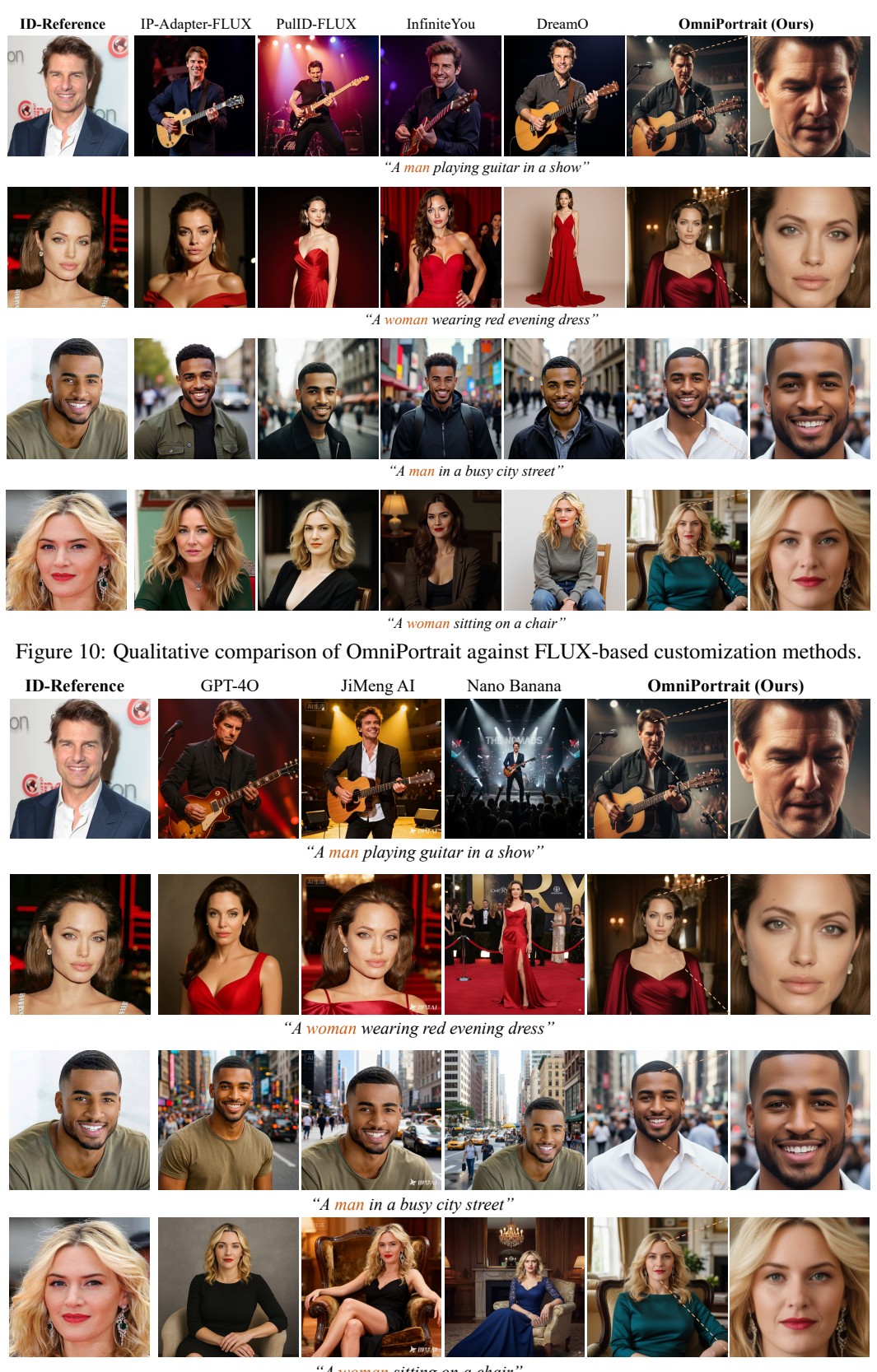

Figure 10: Qualitative comparison of OmniPortrait against FLUX-based customization methods.

Figure 11: Qualitative comparison of OmniPortrait with closed-source image editing models.

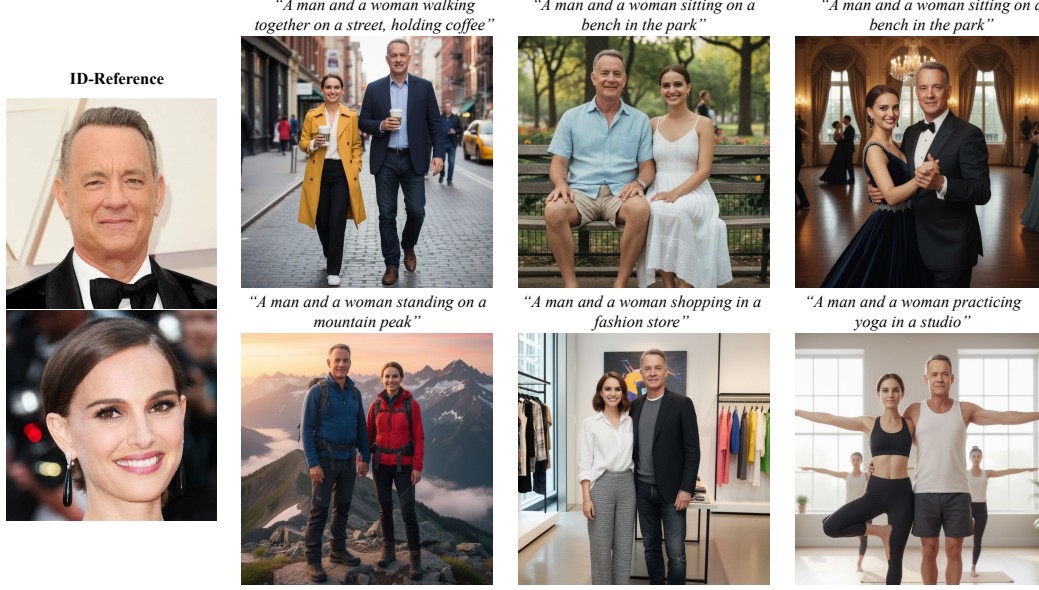

Figure 12: Multi-identity personalization with OmniPortrait: consistent identities and high text–image alignment.

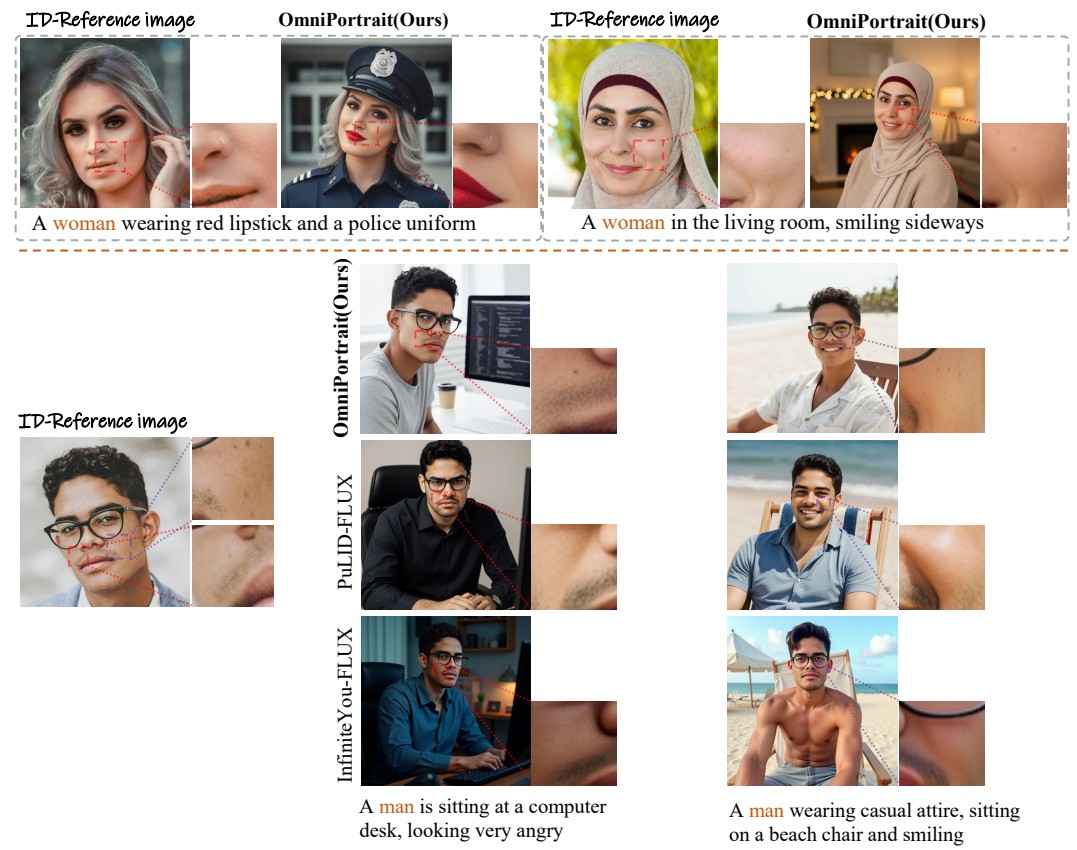

Figure 13: Facial attribute editing via text prompts using OmniPortrait, and a qualitative comparison of facial detail fidelity against other methods.

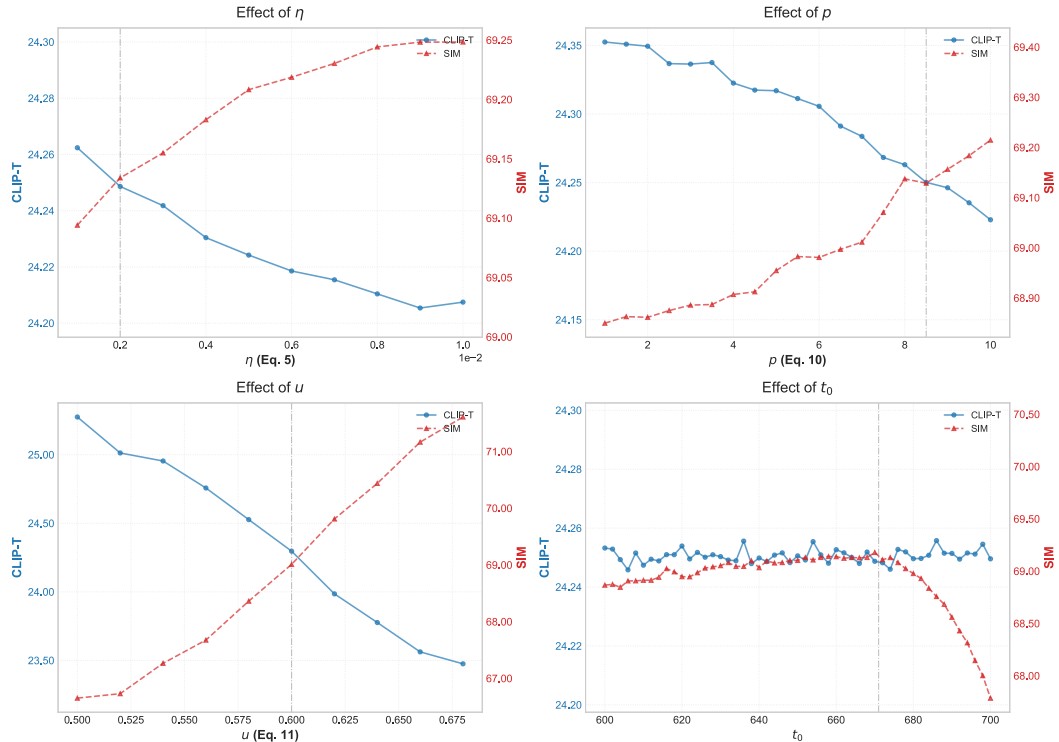

Figure 14: Hyperparameter ablation results. CLIP-T measures text alignment, while SIM measures face similarity.

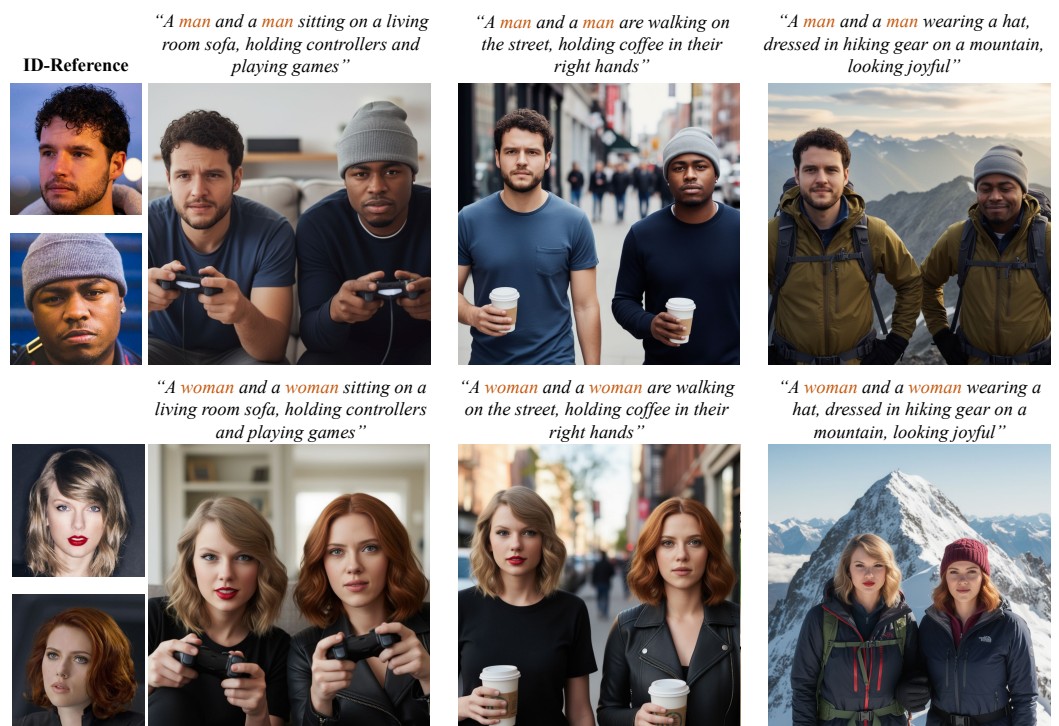

Figure 15: Additional results for multi-identity personalization with OmniPortrait.

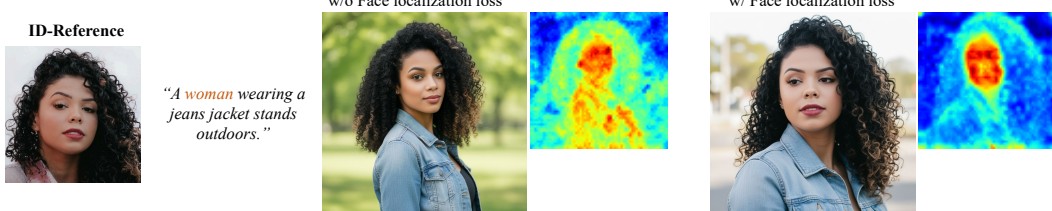

Figure 16: Qualitative examples showing the impact of face localization loss on learned cross-attention maps.

Table 3: Full list of evaluation prompts for subject customization (5 prompts per category).

| Category | Prompt |
|---|---|
| Pose | A woman/man posing like a tourist in front of the Eiffel Tower |
| | A woman/man performing a yoga tree pose |
| | A woman/man sitting cross-legged on the floor, smiling |
| | A woman/man jumping in the air with joy |
| | A woman/man looking back over their shoulder |
| Clothing | A woman/man wearing a police uniform |
| | A woman/man wearing a formal business suit |
| | A woman/man wearing a traditional kimono |
| | A woman/man in a superhero costume, crying disappointedly |
| | A woman/man wearing a chef's white uniform |
| Background | A woman/man on the beach |
| | A woman/man standing in a snowy forest, looking very angry |
| | A woman/man floating in outer space |
| | A woman/man standing in a crowded city street |
| | A woman/man inside an ancient library |
| Action | A woman/man holding a piece of cake |
| | A woman/man playing an acoustic guitar |
| | A woman/man riding a bicycle, with sweat on face. |
| | A woman/man drinking a cup of coffee |
| | A woman/man taking a selfie with a phone |
| Lighting | A woman/man wearing a Christmas hat in a bright scene |
| | A woman/man wearing glasses illuminated by cinematic neon lights |
| | A woman/man standing in the golden hour sunlight |
| | A woman/man in a dark room with dramatic shadows |
| | A woman/man under the moonlight |
| Style | A painting of a woman/man in the style of Vincent Van Gogh |
| | An oil painting of a woman/man |
| | A pencil sketch of a woman/man |
| | A 3D render of a woman/man in Pixar style |
| | A pop art style portrait of a woman/man |

