# OpenReview forum: "OmniPortrait: Fine-Grained Personalized Portrait Synthesis via Pivotal Optimization"
_ICLR.cc/2026/Conference — ICLR 2026 Poster_

### Official Review · Reviewer_Lh8m · 2025-10-26

**Soundness:** 3
**Presentation:** 3
**Contribution:** 2
**Rating:** 4
**Confidence:** 5

**Summary:**

This paper explores the field of personalized image generation. It begins by introducing the Pivot ID Encoder to achieve coarse-grained identity consistency, thus avoiding the degradation of prompt alignment caused by directly fine-tuning the base model. The authors then propose RB-Guidance to enhance fidelity during inference. This approach leverages the correlation between the denoised latent features of the generated image and the reference image, improving the fidelity of facial features. Additionally, the paper presents the construction and curation of the large-scale dataset OmniPortrait-1M, which will contribute to further advancements in the community.

**Strengths:**

- The proposed method can be extended to multi-ID personalization without requiring multi-ID training data.
- The paper is well-written and easy to follow.
- The ablation study is thorough and effectively demonstrates the validity of each module.
- The introduction of the OmniPortrait-1M dataset is a valuable contribution.

**Weaknesses:**

- The paper claims that current state-of-the-art methods suffer from two main issues: low facial fidelity and failure to maintain prompt alignment. However, the methods demonstrated (InstantID, PhotoMaker, FastComposer) are outdated. In fact, more advanced approaches such as ConsistentID, Pulid, and InfiniteYou have already addressed these issues, making the motivation less academically compelling.
- 1）The operation of associating visual embeddings with text embeddings in the Pivot ID Encoder has already been widely adopted, for example, in PhotoMaker. Similarly, the idea of face localization loss is also present in FastComposer.   2）The insight behind RB-Guidance—utilizing diffusion feature matching to improve facial fidelity—has been extensively explored by other methods, such as Visual Persona and Consistory.
- The base model used in the paper, based on the U-Net architecture, is outdated. Currently, competitive base models are using the DiT architecture, and the effectiveness of the proposed method on DiT remains untested.
- The methods compared in the paper are outdated, while results comparing to more advanced methods are relegated to the appendix. Furthermore, only qualitative results are provided, without any quantitative experiments.
-  When using InstantID, the reference image should not be cropped. Instead, the original reference image should be provided.

**Questions:**

- This method does not propose a solution to the multi-person identity confusion problem but instead relies on the semantics of the prompt for distinction, e.g., “a man and a woman.” How would the method perform if the prompt were “two men” or “a man and a man”? Would it still function properly?
- On line 377, the authors state that previous methods often produce results resembling a “copy-and-paste” of the reference image, which is expected, since during training, the reference face and the ground truth face are always aligned. However, this method does not appear to make any substantial improvements to this issue during training. Why does this method avoid this problem in the qualitative experiments?
- The authors claim to have constructed and released OmniPortrait-1M, a large-scale, high-quality multimodal face dataset. However, I don't find access provided in the manuscript or appendix. Will the dataset be released upon acceptance?

---

> ### Author Response · Authors · 2025-11-19
> **Responses to the Reviewer Lh8m [1/3]**
>
> We sincerely appreciate your thoughtful comments and your recognition that our paper is "well-written and easy to follow" with a thorough ablation study. We are also pleased that you noted our method "can be extended to multi-ID personalization" and that the OmniPortrait-1M dataset serves as a valuable contribution. Below, we provide detailed responses to your questions.
>
> **Q1: The paper claims that current state-of-the-art methods suffer from two main issues: low facial fidelity and failure to maintain prompt alignment. However, the methods demonstrated (InstantID, PhotoMaker, FastComposer) are outdated. In fact, more advanced approaches such as ConsistentID, Pulid, and InfiniteYou have already addressed these issues, making the motivation less academically compelling.**
>
> **A1:** Thanks for your insightful comments. While simultaneously enhancing face fidelity and text alignment remains a long-standing challenge in personalized portrait generation, our findings indicate that existing methods—including InstantID, PhotoMaker, Pulid, and InfiniteYou—all struggle to retain authentic facial details such as beauty marks and subtle wrinkles, which are essential for achieving photorealism, as shown in Fig. 1 and Fig. 13 (page 17). Motivated by these observations, we propose a method capable of achieving high-fidelity facial detail preservation without compromising text alignment.
>
> **Q2: The operation of associating visual embeddings with text embeddings in the Pivot ID Encoder has already been widely adopted and utilizing diffusion feature matching to improve facial fidelity has been extensively explored by other methods.**
>
> **A2:**  Thanks for your insightful comments. We would like to emphasize that our core contribution lies in the proposed pivotal optimization paradigm. This coarse-to-fine approach achieves high-fidelity personalized portrait synthesis without compromising text alignment, fundamentally distinguishing our work from existing methods that rely solely on improving encoders and base models.
>
> Furthermore, to the best of our knowledge, we are the first to utilize diffusion feature matching for reference-based face identity customization. In comparison, Visual Persona is an encoder-based method operating without additional guidance, while ConsiStory does not support reference-based generation.
>
> **Q3: The base model used in the paper, based on the U-Net architecture, is outdated. Currently, competitive base models are using the DiT architecture, and the effectiveness of the proposed method on DiT remains untested.**
>
> **A3:** Thanks for your valuable comments. We explain it to you from the following three aspects:
> - Our findings suggest that despite leveraging advanced base models and ID encoders, existing methods still struggle to balance text alignment and identity preservation. Therefore, we aim to break away from the prevailing paradigm that relies solely on upgrading base models and encoders. Instead, we propose a pivotal optimization paradigm to generate customized portraits in a coarse-to-fine manner. Experimental results demonstrate that our approach outperforms state-of-the-art methods (based on both SDXL and Flux), further validating the effectiveness of the proposed paradigm.
> - Our core idea is pivotal optimization, which necessitates a feature space capable of maintaining accurate visual correspondence. During our implementation, we observed that the inter-layer features of UNet-based architectures (e.g., SD&SDXL) exhibit this essential property.
> - It is worth noting that our paradigm is adaptable. While our experiments confirmed that the inter-layer features of DiTs lack inherent visual correspondence, we discovered that the Adaptive Layer Normalization (AdaLN) components in DiTs do possess this property and exhibit superior feature matching capabilities. This observation is corroborated by the recent work “Unleashing Diffusion Transformers for Visual Correspondence by Modulating Massive Activations.” We have already conducted preliminary experiments yielding highly promising intermediate results, and we plan to release a FLUX-based model upon publication.

---

> ### Author Response · Authors · 2025-11-19
> **Responses to the Reviewer Lh8m [2/3]**
>
> **Q4: The methods compared in the paper are outdated, while results comparing to more advanced methods are relegated to the appendix. Furthermore, only qualitative results are provided, without any quantitative experiments.**
>
> **A4:** Our primary rationale for prioritizing comparisons with methods using similar base architectures was to more accurately demonstrate the effectiveness of our proposed pivotal optimization paradigm and RB-guidance. Due to space limitations, we placed the comparison with DiT-based methods in the Appendix.
>
> Following your advice, we present quantitative comparisons in the table below. When combined with qualitative results shown in Fig. 10 and Fig. 13, it is evident that while DiT-based methods show improvements in text alignment and face fidelity, they still fall short of our method in preserving fine-grained facial details."
>
> | Methods | BLIP | CLIP-T | CLIP-I | SIM |
> | :--- | :---: | :---: | :---: | :---: |
> | PuLID-FLUX | 79.85 | 23.90 | 72.45 | 68.55 |
> | InfiniteYou | 78.92 | 24.05 | 71.15 | 67.79 |
> | DreamO | **80.33** | 24.17 | 72.88 | 68.12 |
> | OmniPortrait | 80.24 | **24.25** | **73.08** | **69.13** |
>
> **Q5: When using InstantID, the reference image should not be cropped. Instead, the original reference image should be provided.**
>
> **A5:** Thanks for your valuable advice. Following your suggestion, we have aligned the pose condition of InstantID with our method to facilitate comparison. This correction has been made in the revised version.

---

> ### Author Response · Authors · 2025-11-19
> **Responses to the Reviewer Lh8m [3/3]**
>
> **Q6: This method does not propose a solution to the multi-person identity confusion problem but instead relies on the semantics of the prompt for distinction, e.g., “a man and a woman.” How would the method perform if the prompt were “two men” or “a man and a man”? Would it still function properly?**
>
> **A6:** Thanks for your insightful comments. In Section 3.3, we introduce Background Gradient Masking to constrain the guidance of each identity to specific regions. Consequently, personalized portrait synthesis for multi-person scenarios requires no additional designs, as each identity is treated simply as the masked-out background relative to the others. Following your advice, we present additional multi-person customization results in Figure 15 on page 18, featuring cases with two men and two women, respectively.
>
> **Q7: On line 377, the authors state that previous methods often produce results resembling a “copy-and-paste” of the reference image, which is expected, since during training, the reference face and the ground truth face are always aligned. However, this method does not appear to make any substantial improvements to this issue during training. Why does this method avoid this problem in the qualitative experiments?**
>
> **A7:** Thanks for your detailed review of our manuscript.  The "copy-and-paste" phenomenon observed in previous methods is primarily attributed to two factors. First, they typically train the entire denoising network (e.g., FastComposer). Second, insufficient training data leads models to learn simple mappings, resulting in overfitting.
>
> In contrast, our proposed Pivotal ID Encoder fully preserves the pre-trained parameters of the base model. While this comes at the cost of initially lower face similarity, it effectively retains the model's rich priors. The face similarity is subsequently boosted by RB-Guidance in an on-the-fly manner. Furthermore, data scarcity predisposes existing methods to overfit on easily accessible portrait data, such as celebrities. Conversely, our proposed OmniPortrait-1M is a diverse dataset containing a vast number of distinct identities.
>
> **Q8: The authors claim to have constructed and released OmniPortrait-1M, a large-scale, high-quality multimodal face dataset. However, I don't find access provided in the manuscript or appendix. Will the dataset be released upon acceptance?**
>
> **A8:** We appreciate your interest in our dataset. We will release OmniPortrait-1M upon the paper's acceptance to facilitate further advancements in the community.

---

> > ### Comment · Reviewer_Lh8m · 2025-11-22
> >
> > We appreciate the authors' detailed response to our comments and the effort put into the rebuttal, including the addition of new quantitative results. However, after reviewing the arguments, several key concerns regarding the novelty, methodology, and generalizability of the proposed work remain unresolved.
> >
> > ### A1
> >
> > I acknowledge the authors' demonstration, supported by qualitative experiments, that existing state-of-the-art methods struggle to retain authentic fine-grained facial details. This high-fidelity preservation appears to be a notable strength of the proposed method.
> >
> > ### A2
> >
> > I maintain that the essence of the proposed pivotal optimization paradigm is the utilization of **diffusion feature matching**. While the method of introducing this through a gradient-based approach may be more refined than direct feature injection (e.g., using cosine similarity as in consiStory), the underlying mechanism remains fundamentally similar. Thus, I still consider the contribution to be **incremental** rather than a paradigm shift.
> >
> > The authors state that "ConsiStory does not support reference-based generation." This is incorrect. Please refer to Section 4.4 of the ConsiStory paper, which discusses its capability for reference-based identity injection. This inaccurate comparison further weakens the claim of novel application.
> >
> > ### A3
> >
> > The authors state that their method requires a specific feature space for accurate visual correspondence, a property currently found only in outdated UNet-based architectures (e.g., SD, SDXL), and is incompatible with state-of-the-art models built on the DiT architecture (e.g., Flux).  Advanced DiT-based models offer superior capabilities in terms of text-editing ability, aesthetic quality, and physical plausibility. The method's severe reliance on a **specialized feature space exclusively found in an outdated architecture**, which is being increasingly phased out in the image generation domain, significantly **undermines the generalizability and long-term academic relevance** of this work. The inability to adapt to the most powerful, cutting-edge foundation models is a critical limitation.
> >
> >
> >
> > ### A4
> >
> > I appreciate the inclusion of new quantitative experiments comparing the proposed method with DiT-based approaches. This new table effectively resolves the initial concern and robustly demonstrates the method's effectiveness in terms of text alignment and face fidelity compared to the new baselines.
> >
> > ### A6
> >
> > I am still confused because the Background Gradient Masking is generated dynamically at inference time. For a prompt like 'two men,' the word 'men' would naturally map to two indistinguishable face masks. Then, how is identity differentiation achieved?
> >
> > ### A7
> >
> > I find the first explanation less compelling. Many established and successful methods (e.g., IP-Adapter, InfiniteYou) also do not involve training the entire denoising network. Therefore, I suspect this favorable characteristic is more likely attributed to the diversity and scale of the proposed OmniPortrait-1M dataset rather than the architectural design choices during training.
> >
> > ### A8
> >
> > I acknowledge the authors' commitment to releasing the OmniPortrait-1M dataset upon acceptance. I think this large-scale, high-quality dataset is a **core contribution** of the work, and I will closely monitor its public release as promised.
> >
> >
> >
> > Therefore, based on the assessment above, we will maintain our original rating.

---

> > > ### Author Response · Authors · 2025-11-25
> > > **Responses to the Reviewer Lh8m [1/2]**
> > >
> > > Thank you for your additional questions. We are glad that the concerns in **Q1**, **Q4**, and **Q8** were resolved in our previous response. Below, we provide detailed responses to your new questions.
> > >
> > > ### **Response to the Follow-up on A2:**
> > > We sincerely thank you for your valuable feedback. However, we respectfully suggest that there may be a misunderstanding regarding our core contributions. We would like to clarify this from the following three perspectives:
> > >
> > > - Novelty of the Paradigm: The primary contribution of our proposed pivotal optimization paradigm extends beyond merely introducing a feature injection technique. Instead, our key innovation lies in analyzing the inherent limitations of existing encoder-based ID customization methods. Based on this analysis, we propose a coarse-to-fine strategy that first identifies a "pivot" and subsequently performs detailed optimization to achieve superior performance.
> > >
> > > - Distinction from Direct Feature Injection: Our proposed RB-Guidance is fundamentally distinct from direct feature injection (e.g., ConsiStory) in two critical aspects:
> > > First, the self-attention feature injection used in ConsiStory is restricted to interactions between images within the same batch. In contrast, our approach allows for arbitrary reference images provided by the user.
> > > Second, as noted in the ConsiStory paper, direct feature replacement tends to force structural similarity across images, thereby reducing the variation of the target images. Conversely, our gradient-based RB-Guidance reinforces key semantic features without compromising generative diversity.
> > >
> > > - Empirical Evidence: To further clarify the essential difference, we conducted an additional experiment where we replaced RB-Guidance with a direct feature replacement mechanism. The results, presented in the table below, demonstrate that a  reference-based replacement strategy fails to function effectively in our scenario.
> > > | Methods | BLIP | CLIP-T | CLIP-I | SIM |
> > > | :--- | :---: | :---: | :---: | :---: |
> > > | Baseline (OmniPortrait without RB-Guidance) | 81.05 | **24.88** | 66.10 | 63.87 |
> > > | Baseline + Reference-based Attention | 65.90 | 19.29 | 70.49 | 65.16 |
> > > | Baseline + RB-Guidance | **80.24** | 24.25 | **73.08** | **69.13** |
> > >
> > > Thank you for your follow-up question, and we apologize for any confusion caused by our earlier response. When we stated that "ConsiStory does not support reference-based generation," we specifically meant that ConsiStory is incapable of utilizing an arbitrary, user-provided image as a reference. As described in Section 4.4 of the ConsiStory paper, a user must first generate an anchor image containing the desired subject via text, and only then "reuse the same subjects in novel scenes by creating a new batch where the same prompts and seeds are used to re-create the anchor images, but the non-anchor prompts have changed."
> > >
> > > In contrast, our task is designed to work with any given reference image, not just those synthetically generated by the model itself. In real-world applications, users typically want to use their own photos (ID images) as references for customized generation, yet it is nearly impossible to generate an image that accurately preserves a specific person's identity solely through text prompts. This implies a fundamental difference between our approach and the direct feature injection method used in ConsiStory.
> > >
> > > ### **Response to the Follow-up on A3:**
> > > Thank you for your insightful response, and we apologize for the confusion in our earlier reply. In our previous response, we primarily focused on explaining the rationale behind our choice of the U-Net architecture, specifically noting its unique feature space properties and the fact that our proposed paradigm already outperforms existing DiT-based methods. However, this does not imply that our proposed pivot optimization method cannot be extended to DiT-based models. While our experiments confirmed that the inter-layer features of DiTs lack inherent visual correspondence, we discovered that the Adaptive Layer Normalization (AdaLN) components in DiTs do possess this property and exhibit superior feature matching capabilities. This observation is corroborated by the recent work “Unleashing Diffusion Transformers for Visual Correspondence by Modulating Massive Activations.” We have already conducted preliminary experiments yielding highly promising intermediate results, and we plan to release a FLUX-based model upon publication.

---

> > > ### Author Response · Authors · 2025-11-25
> > > **Responses to the Reviewer Lh8m [2/2]**
> > >
> > > ### **Response to the Follow-up on A6:**
> > > Thank you for the further question. In our method, nouns representing the subject, such as "woman" or "man," effectively serve as trigger words. This is because we concatenate the facial features encoded by the Pivot ID Encoder with the corresponding nouns to derive the mixed pivot embeddings. Consequently, in multi-person customization scenarios, the correct approach is to map each distinct identity reference to a specific noun, rather than using collective terms like "two women" or "two men."
> > >
> > > ### **Response to the Follow-up on A7:**
> > > Thank you for your further comment. Regarding our initial statement in Line 377, we observed that only specific methods, such as FastComposer and IP-Adapter, produce results resembling a “copy-and-paste” of the reference image, whereas other methods like InfiniteYou do not exhibit this behavior. FastComposer fine-tunes the entire denoising network, while IP-Adapter injects additional cross-attention modules; both approaches tend to degrade the priors of the base model. Our experiments corroborate this finding: when we followed the IP-Adapter strategy by training additional cross-attention keys and values, we did achieve improved identity fidelity, but this came at the cost of reduced text alignment, as shown in the table below. This trade-off significantly limits the variations in the generated images supported by the model.
> > > | Methods | BLIP | CLIP-T | CLIP-I | SIM |
> > > | :--- | :---: | :---: | :---: | :---: |
> > > | OmniPortrait + Frozen Linear Layer | 76.68 | 21.20 | 49.89 | 33.14 |
> > > | OmniPortrait | **80.24** | **24.25** | 73.08 | 69.13 |
> > > | OmniPortrait + Learnable Cross-Attn K/V | 77.32 | 22.77 | **73.11** | **70.94** |
> > >
> > > Once more, we sincerely thank you for all the comments and very useful feedback. We think that we have addressed all the questions in depth. If the reviewer has any additional questions, please let us know, and we will be more than happy to answer them.

---

> ### Author Response · Authors · 2025-11-28
>
> Dear Reviewer
>
> Could we kindly inquire if the responses have satisfactorily tackled your concerns, or if there is a need for further experiment and visualization? Your commitment to reviewing our work is immensely appreciated. We sincerely thank you for your prompt and insightful review of our paper. Your comment is immensely appreciated and undoubtedly helps improve the quality of our work.

---

### Official Review · Reviewer_V3R8 · 2025-10-30

**Soundness:** 3
**Presentation:** 3
**Contribution:** 2
**Rating:** 6
**Confidence:** 3

**Summary:**

This work proposes method called OmniPortrait to generate photo-realistic image while preserve identity consistency and achieve high alignment with text prompt by introducing pivotal optimization and reference-based guidance (RB-Guidance). It also supports multi-identity customization.

**Strengths:**

1. Only Pivot ID Encoder and linear protection layer are needed for training which makes OmniPortrait more efficient than finetuning UNet model.
2. The proposed RB-Guidance enable OmniPortrait to have better ID-preserving generation.
3. Authors have conducted various experiments to verify the effectiveness of the method.

**Weaknesses:**

1. The size of evaluation dataset is too small, only 50 reference images are used. Moreover, the content of 30 text prompts used are not clear. Based some samples in Figure 4, it seems the prompts mostly descript general object rather than descript facial attribute itself.

2. Authors claim that OmniPortrait have better
fine-grained identity details. However, it seems no relevant experiments support this. For example, authors should measure how facial attributes changes in generated face images according to text prompts that descript the wanted facial attributes.

3. Author use RB-Guidance to optimize the energy function. However, how energy function associated with ID-preservation generation is under explained.

**Questions:**

1. Author mentioned they utilized SD and SDXL in Sec. 4.2. However, which base model are used in each experiment?
2. For measuring ID-preservation, why authors didn’t measure cosine distance between reference and generated faces images? It will be more convinced if measuring recognition accuracy among different face recognition models.
Other questions see in weakness.

---

> ### Author Response · Authors · 2025-11-19
> **Responses to the Reviewer V3R8 [1/2]**
>
> We sincerely appreciate your thoughtful comments, especially noting that our method is "more efficient than finetuning UNet model" and that we have conducted "various experiments to verify the effectiveness of the method." We are also glad you recognized that the proposed RB-Guidance leads to "better ID-preserving generation." Below, we provide detailed responses to your questions.
>
> **Q1: The size of evaluation dataset is too small, only 50 reference images are used. Moreover, the content of 30 text prompts used are not clear. Based some samples in Figure 4, it seems the prompts mostly descript general object rather than descript facial attribute itself.**
>
> **A1:** Thanks for your valuable advice and we acknowledge that a larger evaluation dataset would indeed be more persuasive. However, for all personalized portrait generation methods, a balance must be struck between evaluation scope and experimental efficiency. To contextualize our setting, we provide a horizontal comparison of the evaluation dataset sizes used in baseline papers, as shown in the table below.
> | Evaluation | FastComposer | PhotoMaker | InfiniteYou | OmniPortrait |
> | :--- | :---: | :---: | :---: | :---: |
> | No. of Identities | 15 | 25 | 15 | 50 |
> | Prompts per Identity | 30 | 40 | ~ | 30 |
> | Total Images | 450 | 1000 | 1497 | 1500 |
>
> Furthermore, we have included the full list of prompts in Table 3 on page 19 of the Appendix in the revised version. Note that facial attribute modification was taken into account during our evaluation, with 20% of the text prompts involving facial attribute editing.
>
> **Q2:Authors claim that OmniPortrait have better fine-grained identity details. However, it seems no relevant experiments support this. For example, authors should measure how facial attributes changes in generated face images according to text prompts that descript the wanted facial attributes.**
>
> **A2:** Thanks for your detailed review of our manuscript. Specifically, the 'identity details' we refer to are fine-grained features, such as beauty marks and subtle wrinkles. We consider these details essential for the realism of customized portraits. As demonstrated in Figure 13 on page 17, unlike existing methods, our proposed pivotal optimization effectively preserves fine-grained identity details under text-driven facial attribute control.

---

> ### Author Response · Authors · 2025-11-19
> **Responses to the Reviewer V3R8 [2/2]**
>
> **Q3: Author use RB-Guidance to optimize the energy function. However, how energy function associated with ID-preservation generation is under explained.**
>
> **A3:** Thanks for your valuable comments. We explain it to you from the following two aspects:
> - Fundamentally, personalized portrait generation involves imposing controllability on the diffusion model's generation process. Currently, two primary paradigms exist for this control: Classifier-Free Guidance (CFG) and energy function guidance.
> - As detailed in Section 3.3, we leverage diffusion feature correspondence to formulate a similarity metric (Eq. 9) between the reference ID image and the generated image in the feature dimension, capturing both facial ID and details. By adopting the maximization of this similarity as our optimization objective, we backpropagate gradients to the noisy latent $z_t$ at inference timestep $t$, thereby steering $z_t$ towards the imposed control conditions via Eq. 4 during the guidance period.
>
> **Q4: Author mentioned they utilized SD and SDXL in Sec. 4.2. However, which base model are used in each experiment?**
>
> **A4:** Thanks for your detailed review of our manuscript. In all experiments, SDXL serves as the base model unless stated otherwise and we have explicitly added this clarification to the revision.
>
> **Q5: For measuring ID-preservation, why authors didn’t measure cosine distance between reference and generated faces images? It will be more convinced if measuring recognition accuracy among different face recognition models.**
>
> **A5:** Thanks for your insightful advice. To clarify, we indeed employed cosine similarity for the calculation of ID-preservation, and we have updated the manuscript to make this explicit. Furthermore, following your suggestion, we extended our evaluation to include results from multiple distinct face recognition models, as shown in the table below.
> | Face Sim. | DreamBooth | Textual Inversion | FastComposer | PhotoMaker | InstantID | IP-Adapter | OmniPortrait |
> | :--- | :---: | :---: | :---: | :---: | :---: | :---: | :---: |
> | FaceNet | 60.07 | 58.32 | 64.97 | 62.68 | 68.35 | 64.19 | **69.13** |
> | CurricularFace | 63.54 | 60.12 | 67.82 | 65.09 | 73.44 | 62.93 | **74.81** |
> | ArcFace | 66.21 | 62.45 | 70.33 | 67.88 | 75.92 | 69.15 | **77.50** |

---

> ### Author Response · Authors · 2025-11-28
>
> Dear Reviewer
>
> Could we kindly inquire if the responses have satisfactorily tackled your concerns, or if there is a need for further experiment and visualization? Your commitment to reviewing our work is immensely appreciated. We sincerely thank you for your prompt and insightful review of our paper.

---

### Official Review · Reviewer_iUtW · 2025-11-01

**Soundness:** 3
**Presentation:** 3
**Contribution:** 3
**Rating:** 6
**Confidence:** 4

**Summary:**

This paper introduces a personalized portrait generation framework based on pivotal optimization, i.e., combining the facial representation with a specific text token and optimizing it to preserve the identity. Specifically, the proposed method consists of two parts. First, an efficient pivot ID encoder is designed to generate the combined representation, which is learned by a face localization loss calculated between cross-attention maps and facial segmentation masks. Second, during the inference, a pivotal optimization objective is proposed to capture the feature correspondence between the reference image and the generated image. Moreover, a large-scale dataset is collected to train the proposed ID encoder. The experimental results on 50 reference images sampled from the CelebA-HQ and FFHQ datasets show that the proposed method outperforms a few baselines.

**Strengths:**

+ The proposed pivot ID encoder is simple (feature concatenation followed by a linear layer) and efficient. And from the quantitative results, it improves the base model significantly.

+ The designed facial localization loss and the reference-based guidance loss are reasonable and can be combined with other models seamlessly.

+ Compared with the baselines, the proposed method balances text following and identity preservation better.

+ This paper is written well and conveys its idea clearly.

**Weaknesses:**

1. There are many hyperparameters (Eq. (3), (5), (6), (10), and (11)) in the proposed method, of which the settings seem to overfit the dataset (e.g., $t_0 = 671$) and require in-depth analysis.

2. According to the ablation study (Table 2), the proposed face localization loss contributes most to the performance gains of the proposed method. However, it also lacks a comprehensive analysis, e.g., the qualitative examples of learned cross-attention maps, comparisons with other localization losses, and the selection of denoising layers for learning.

3. The base models of the proposed method and the baselines are out-of-date (e.g., SDXL). As a few qualitative results of some FLUX-based methods are reported in the Appendix, why not implement the proposed method based on FLUX for comparisons directly in the main paper?

**Questions:**

Q1: Among the baselines, are they trained on the proposed OmniPortrait-1M dataset as well (at least the key components of one baseline)? If not, the comparisons may be unfair.

Q2: Will the proposed dataset be released?

Q3: How does the proposed method perform in the wild (e.g., non-celebrity)?

Miscs.:
+ "PulID-FLUX" in Figure 10 should be "PuLID-FLUX".

**Details Of Ethics Concerns:**

The proposed method is designed for personalized image generation, which may be used maliciously.

---

> ### Author Response · Authors · 2025-11-19
> **Responses to the Reviewer iUtW [1/2]**
>
> We sincerely appreciate your thoughtful comments and your recognition that our proposed method "balances text following and identity preservation better" compared with baselines. We are also encouraged by your observation that the Pivot ID encoder is "simple and efficient" and "improves the base model significantly." while being "written well." Below, we provide detailed responses to your questions.
>
> **Q1: There are many hyperparameters (Eq. (3), (5), (6), (10), and (11)) in the proposed method, of which the settings seem to overfit the dataset (e.g., t0=671) and require in-depth analysis.**
>
> **A1:** Thanks for your detailed review of our manuscript. The CFG weight ($w=7.5$) in Eq. 3 and the morphological operation parameters in Eq. 6 follow widely used community default setting. For other hyperparameters—specifically $\eta$ (Eq. 5), $p$ (Eq. 10), and $u$ (Eq. 11), $t_0$ —values were determined via parameter search, as detailed in Figure 14 on page 18. These selections were made to ensure a trade-off that balances face similarity with text alignment. In addition, we analyzed the choice of hyperparameters in Appendix A.4 of the revised version.
>
> **Q2：According to the ablation study (Table 2), the proposed face localization loss contributes most to the performance gains of the proposed method. However, it also lacks a comprehensive analysis, e.g., the qualitative examples of learned cross-attention maps, comparisons with other localization losses, and the selection of denoising layers for learning.**
>
> **A2:**  Thanks for your valuable advice. We illustrate the qualitative ablation of the facial localization loss in Figure 16 on page 19. The upper-right corner displays the inference-time cross-attention map derived from mixed pivot embeddings, which enables accurate localization of the generated face. Additionally, we attempted to vary the training configuration by training fewer or more modules. As shown in the following table, such modifications led to performance degradation.
> | Methods | BLIP | CLIP-T | CLIP-I | SIM |
> | :--- | :---: | :---: | :---: | :---: |
> | OmniPortrait + Frozen Linear Layer | 76.68 | 21.20 | 49.89 | 33.14 |
> | OmniPortrait | **80.24** | **24.25** | 73.08 | 69.13 |
> | OmniPortrait + Learnable Cross-Attn K/V | 77.32 | 22.77 | **73.11** | **70.94** |

---

> ### Author Response · Authors · 2025-11-19
> **Responses to the Reviewer iUtW [2/2]**
>
> **Q3: The base models of the proposed method and the baselines are out-of-date (e.g., SDXL). As a few qualitative results of some FLUX-based methods are reported in the Appendix, why not implement the proposed method based on FLUX for comparisons directly in the main paper?**
>
> **A3:** Thanks for your insightful comments. We explain it to you from the following three aspects:
> - Our core idea is pivotal optimization, which necessitates a feature space capable of maintaining accurate visual correspondence. During our implementation, we observed that the inter-layer features of UNet-based architectures (e.g., SD and SDXL) exhibit this essential property.
> -  Our experimental findings suggest that the performance limitations of existing portrait customization methods are not primarily due to the capabilities of the base model. Despite exploring various combinations of encoders and base models, existing approaches still struggle to balance text following and identity preservation. In contrast, our proposed coarse-to-fine paradigm outperforms SOTA methods (based on both SDXL and Flux) without relying heavily on a stronger base model, which further underscores the intrinsic effectiveness of our proposed paradigm.
> - It is worth noting that our paradigm is adaptable. While our experiments confirmed that the inter-layer features of DiTs lack inherent visual correspondence, we discovered that the Adaptive Layer Normalization (AdaLN) components in DiTs do possess this property and exhibit superior feature matching capabilities. This observation is corroborated by the recent work “Unleashing Diffusion Transformers for Visual Correspondence by Modulating Massive Activations.” We have already conducted preliminary experiments yielding highly promising intermediate results, and we plan to release a FLUX-based model upon publication.
>
> **Q4: Among the baselines, are they trained on the proposed OmniPortrait-1M dataset as well (at least the key components of one baseline)? If not, the comparisons may be unfair.**
>
> **A4:** Since all baseline methods utilize unreleased private datasets, we retrained the two baselines with accessible training code from scratch on our proposed OmniPortrait-1M dataset. The results are shown in the following table:
> | Methods | BLIP | CLIP-T | CLIP-I | SIM |
> | :--- | :---: | :---: | :---: | :---: |
> | InstantID | 78.21 | 22.26 | 73.03 | 68.35 |
> | InstantID + OmniPortrait-1M | 79.40 | 23.46 | 72.98 | 68.52 |
> | IP-Adapter | 79.33 | 23.93 | 68.23 | 64.19 |
> | IP-Adapter + OmniPortrait-1M | 80.16 | **24.28** | 70.82 | 67.55 |
> | OmniPortrait | **80.24** | 24.25 | **73.08** | **69.13** |
>
>
> **Q5: Will the proposed dataset be released?**
>
> **A5:** Thank you for your interest in our proposed dataset. As noted in **Q4**, the lack of public datasets poses a significant challenge for benchmarking methods under a unified setting. This is why the OmniPortrait-1M dataset one of the core contributions of our paper. We pledge to release the dataset publicly immediately upon acceptance, anticipating that it will become a valuable asset for future community research.
>
> **Q6: How does the proposed method perform in the wild (e.g., non-celebrity)?**
>
> **A6:** Thanks for your insightful comments. We present results for non-celebrities in Fig. 13 on page 17 and Fig. 15 (first row) on page 18, demonstrating that our method achieves robust performance in in-the-wild scenarios. Indeed, the sufficient diversity of our proposed dataset underpins the model's strong generalization capabilities.
>
> **Miscs.:** Thanks for your detailed review of our manuscript, and we have made the corrections in the revised version.

---

> ### Author Response · Authors · 2025-11-28
>
> Dear Reviewer
>
> Would it be possible for us to kindly ascertain if the provided responses have satisfactorily tackled any concerns you may have had and if further explanations or clarifications are needed? Your generous investment of time and effort in the evaluation of our work is truly commendable. We extend our heartfelt gratitude for your insightful commentary and the considerable time you have devoted to reviewing our paper.

---

### Official Review · Reviewer_ju3a · 2025-11-01

**Soundness:** 4
**Presentation:** 4
**Contribution:** 4
**Rating:** 6
**Confidence:** 3

**Summary:**

This paper introduces fine-grained personalized portrait synthesis system upon diffusion models. To mitigate the shortcoming of existing models, it propose pivot-based optimization strategies, which ensures the identity of reference image and editability of input prompt.

**Strengths:**

- High fidelity with editability: The proposed system demonstrate pretty plausible outcomes while preserving input fidelity with prompt-based editability with reduced trade-off problem between them.
- Compatibility and extendability: This framework is basically plug-and-play approach with high compatibility. Moreover, some additional results including multi-identity injection showed the significant extendability of the proposed method.

**Weaknesses:**

- Advantage of the proposed pivotal optimization: For reference-based editing, there are several well-known technics in diffusion models such as cross-attention, attention map swapping, style injection via AdaIN and so on. Within the proposed RB-Guidance, there is no comparison about aforementioned approaches before proposing the pivotal-optimization. It is imperative to provide and discuss some limitations of existing guidance manner on reference path and how the proposed one is useful.
- Effects of Pivot ID Encoder: The authors propose a new trainable encoder to extract facial identity using CLIP-based model. However, it is common to adopt identity embedding models such as ArcFace, InsightFace to provide identity vectors within denoising process using cross attention module, etc. It is wondered the proposed Pivot ID Encoder offers more useful information in entire process, compared to off-the-shelf identity extractor.

**Questions:**

- Without pivotal optimization, is there any degradation when adopting existing manner like cross-attention or attention map swapping for reference path?
- Before validating the Pivot ID Encoder, how do off-the-shelf identity extractors work compared to the proposed one?
- It is recommended to discuss the diversity on prompt-based portrait synthesis in terms of facial expression and appearance. The results presented in the manuscript appear to have a tendency to just copy-and-paste the overall facial structure of the reference image, including the open mouth and smile, while PhotoMaker changed those attributes naturally.

---

> ### Author Response · Authors · 2025-11-19
> **Responses to the Reviewer ju3a [1/2]**
>
> We sincerely appreciate your thoughtful comments, especially noting that we maintain "prompt-based editability with reduced trade-off problem" while ensuring high fidelity. We are also glad that you recognized the "significant extendability of the proposed method" and its "plug-and-play" nature. Below, we provide detailed responses to your questions.
>
> **Q1: Advantage of the proposed pivotal optimization: Without pivotal optimization, is there any degradation when adopting existing manner like cross-attention or attention map swapping for reference path?**
>
> **A1:** Thanks for your thoughtful comments. In our proposed pivotal optimization framework, it is essential to conduct reference-based optimization within a space that exhibits strong visual correspondence.
>
> Following your suggestion, we attempted to replace the proposed RB-Guidance with a visual reference-based attention mechanism, while keeping the Pivot ID Encoder unchanged. Specifically, we first performed DDIM inversion on the reference image, storing the keys and values from the self-attention modules at each timestep ($ t=1 $ to $t=T$) as a memory bank. Subsequently, during the generation process, we replaced the keys and values in the self-attention modules across all layers with those retrieved from the memory bank.
>
> As shown in the table below, we adopt OmniPortrait without RB-Guidance as the baseline. We observed that replacing RB-Guidance not only failed to provide sufficient visual correspondence guidance but also compromised text alignment.
> | Methods | BLIP | CLIP-T | CLIP-I | SIM |
> | :--- | :---: | :---: | :---: | :---: |
> | Baseline (OmniPortrait without RB-Guidance) | 81.05 | **24.88** | 66.10 | 63.87 |
> | Baseline + Reference-based Attention | 65.90 | 19.29 | 70.49 | 65.16 |
> | Baseline + RB-Guidance | **80.24** | 24.25 | **73.08** | **69.13** |

---

> ### Author Response · Authors · 2025-11-19
> **Responses to the Reviewer ju3a [2/2]**
>
> **Q2: Effects of Pivot ID Encoder: Before validating the Pivot ID Encoder, how do off-the-shelf identity extractors work compared to the proposed one?**
>
> **A2:** Thanks for your valuable comments. As shown in the first row of the table below, replacing the proposed Pivot ID Encoder with off-the-shelf identity extractors may not suitable for our pivotal optimization paradigm. We explain it to you from the following two aspects:
> - To ensure plug-and-play capability and efficient training, we freeze the denoiser, rendering the fine-tuning of the Pivot ID Encoder indispensable. Off-the-shelf extractors like ArcFace and AntelopeV2, however, demand specific training conditions and are challenging to optimize jointly with diffusion and face localization losses.
> - Since the text conditioning space of the base model is CLIP-based, employing a CLIP-based Pivot ID Encoder generates mixed pivot embeddings that align more closely with the original semantic space. This alignment is crucial for facilitating accurate facial localization.
>
> Drawing inspiration from your suggestion, we attempt to combine the AntelopeV2 identity extractor with our method. Specifically, we concatenate the AntelopeV2-encoded features with our Pivot ID Encoder and pass them through a linear layer to generate the mixed pivot embeddings, while keeping AntelopeV2 frozen. As shown in the table below, we found that identity consistency improved, accompanied by a degradation in text alignment. Despite this trade-off, the findings indicate that this is a promising avenue for further investigation.
>
>
> | Methods | BLIP | CLIP-T | CLIP-I | SIM |
> | :--- | :--- | :--- | :---| :--- |
> | OmniPortrait - PIE + AntelopeV2 | 74.96 | 21.85 | 48.11 | 54.20 |
> | OmniPortrait | 80.24 | 24.25 | 73.08 | 69.13 |
> | OmniPortrait+AntelopeV2 | 79.20(-1.3%) | 24.01(-1.0%) | 73.27(+0.3%) | 70.50(+2.0%) |
>
>
> **Q3: It is recommended to discuss the diversity on prompt-based portrait synthesis in terms of facial expression and appearance.**
>
> **A3:** Thanks for your insightful advice. Facial editing in personalized portrait generation is indeed a challenging task. However, thanks to our proposed coarse-to-fine approach which disentangles facial structure from details, our method achieves effective facial editing, as illustrated in Fig. 13 on page 17.

---

> ### Author Response · Authors · 2025-11-28
>
> Dear Reviewer
>
> May we kindly inquire if the provided responses have adequately addressed any questions you might have had? If there remains a requirement for further explanations or clarifications? We wish to express our sincere gratitude for your meticulous evaluation and for generously investing a significant amount of your time in reviewing our paper. Your feedback would be greatly valued.

---

### Comment · Area_Chair_5iEY · 2025-11-24
**Please engage into discussion with authors and fellow reviewers**

Dear reviewers,
The authors have already provided their responses. Do they address your concerns?
Please engage into the discussion with authors and fellow reviewers.
Thanks!
Best,
AC

---

### Author Response · Authors · 2025-12-03
**Rebuttal and Revision Summary for Submission 5817**

**Dear Area Chairs,**

We sincerely appreciate your consideration of our submission. We are also grateful to all the reviewers for their time and constructive feedback. Below we summarize the contributions and response to the reviewers' concerns.

### **Summary of Contributions**
1. **Novel Paradigm**: We analyzed the inherent limitations of existing encoder-based ID customization methods and proposed a **pivotal optimization-based framework** for detail-preserving identity-customized portrait synthesis to achieve superior performance.
2. **Reference-Based Guidance**: We designed a **novel test-time optimization framework** based on local diffusion feature matching, which enables **training-free preservation of fine-grained identity details**. As acknowledged by **Reviewer ju3a and Reviewer iUtW**, this module can be seamlessly integrated with other models.
3. **SOTA Performance**: OmniPortrait achieves state-of-the-art performance over existing methods **(both open-source and proprietary)**. Crucially, it effectively mitigates the trade-off between text alignment and ID fidelity in ID customization tasks, a capability appreciated by **all reviewers**.
4. **Large-scale Dataset**: We constructed OmniPortrait-1M, a **large-scale, high-quality multimodal face dataset** that will contribute to further advancements in the community. This contribution was explicitly commended by **Reviewer iUtW and Reviewer Lh8m**.
5. **Compatibility and Extendability**: The core components of OmniPortrait—the pivotal optimization paradigm and RB-Guidance—are highly **compatible plug-and-play** modules. Furthermore, as highlighted by **Reviewer ju3a, Reviewer Lh8m and Reviewer V3R8**, our method **extends to multi-ID customization without additional training**.

### **Summary of Rebuttal and Revision**

- **Advantage of Proposed Components**: We conducted additional quantitative ablation studies to validate the efficacy of the pivotal optimization and Pivot ID Encoder. Furthermore, addressing concerns from **Reviewer ju3a and Reviewer iUtW**, we included a qualitative ablation of the face localization loss in **Figure 16 (Page 19)**.

- **Facial Attribute Editing**: In response to **Reviewer ju3a and Reviewer V3R8**, we provided extensive qualitative results demonstrating that OmniPortrait achieves effective facial editing, as illustrated in **Figure 13 (Page 17)**.

- **Implementation Details**: We have incorporated hyperparameter search results in **Figure 14 (Page 18)** and **Appendix A.4**. Additionally, we clarified the specific details of the base model in the main text, as requested by **Reviewer iUtW and Reviewer V3R8**.

- **Base Model Selection**: We clarified our rationale for selecting U-Net as the base model. Addressing **Reviewer iUtW and Reviewer Lh8m**, we further discussed how our method can be extended to DiT architectures via Adaptive Layer Normalization (AdaLN)."

- **OmniPortrait-1M Dataset**: We trained baseline models on the OmniPortrait-1M dataset and the results indicate a general performance gain for baselines, yet OmniPortrait maintains SOTA performance. We also reaffirmed our commitment to releasing the dataset publicly upon acceptance (addressing **Reviewer iUtW and Reviewer Lh8m**).

- **Performance in the Wild**: We included additional results for non-celebrity subjects in **Figure 13 (Page 17)** and **Figure 15 (first row, Page 18)**, demonstrating that our method achieves robust performance in in-the-wild scenarios (addressing **Reviewer iUtW**).

- **Evaluation Datasets and Metrics**: In response to **Reviewer V3R8**, we provided a comparison with evaluation datasets used in existing customization methods and listed the text prompts in **Table 3 (Page 19)**. Following reviewer suggestions, we performed quantitative evaluations using different face recognition models.

- **Energy Function**: In response to **Reviewer V3R8**, we provided a theoretical explanation of the energy function as a form of classifier guidance, clarifying its underlying rationale from a conditional guidance perspective.

- **Novelty of RB-Guidance**: In response to **Reviewer Lh8m**, we clarified **misunderstandings regarding related works**: Visual Persona is effectively an encoder-based method operating without additional guidance, and ConsiStory does not support reference-based generation. Furthermore, we conducted quantitative comparisons between **RB-Guidance and Direct Feature Injection** to highlight their fundamental differences.

- **Prevention of Overfitting**: Quantitative experiments demonstrate that OmniPortrait’s lightweight training strategy not only ensures plug-and-play capability but also effectively mitigates overfitting (addressing **Reviewer iUtW and Reviewer Lh8m**).

We believe the substantial revisions have thoroughly addressed all reviewer concerns, and we are confident the revised manuscript constitutes a significantly improved contribution. Thanks again for your time and effort!

Best,

Authors

---

### Meta-Review · Area_Chair_QfPn · 2026-01-06

**Summary:**

This paper introduces a dual-stream diffusion framework for identity-preserved portrait synthesis. After the rebuttal, although the scores remained unchanged (4666 → 4666), most concerns were addressed. Reviewer ju3a and iUtW recognized the technical innovation of the pivotal optimization-based framework and its compatibility with other models. All reviewers acknowledged the strong empirical performance, with OmniPortrait achieving state-of-the-art results and effectively balancing text alignment and identity fidelity. Reviewer ju3a, Lh8m, and V3R8 appreciated the plug-and-play nature and extendability of the method, including support for multi-ID customization without extra training. Reviewer iUtW and Lh8m also commended the contribution of the large-scale OmniPortrait-1M dataset for the community. While some concerns from Reviewer Lh8m about the incremental nature of the contribution and the method’s extension to the DiT architecture remain, the authors provided detailed clarifications that distinguish their approach from existing methods and explained the extension via AdaLN.

After a careful assessment of the submission, reviews, response, and the discussion, the AC recommends acceptance. The authors should further improve the manuscript according to reviewer feedback, especially by clarifying the method’s extension to DiT architecture, explicitly addressing the novelty of their approach, and ensuring the timely public release of the OmniPortrait-1M dataset.

**Reviewer Concerns:**

Most concerns were addressed. Reviewer ju3a and iUtW recognized the technical innovation of the pivotal optimization-based framework and its compatibility with other models. All reviewers acknowledged the strong empirical performance, with OmniPortrait achieving state-of-the-art results and effectively balancing text alignment and identity fidelity. Reviewer ju3a, Lh8m, and V3R8 appreciated the plug-and-play nature and extendability of the method, including support for multi-ID customization without extra training. Reviewer iUtW and Lh8m also commended the contribution of the large-scale OmniPortrait-1M dataset for the community.

While some concerns from Reviewer Lh8m about the incremental nature of the contribution and the method’s extension to the DiT architecture remain, the authors provided detailed clarifications that distinguish their approach from existing methods and explained the extension via AdaLN.

**Reviewer Scores:**

The manuscript received initial review scores of 4/6/6/6. After the rebuttal/discussion and before the reset, the score remained unchanged.

Since most concerns from the Reviewers are addressed after the rebuttal (see 'Reviewer Concerns'), I would approximate 4/6/6/8(Reviewer ju3a) as the final score.

---

### Decision · Program_Chairs · 2026-01-26

Accept (Poster)